# PAC-FNO: Parallel-Structured All-Component Fourier Neural Operators for Recognizing Low-Quality Images

**Jinsung Jeon[1], Hyundong Jin[1], Jonghyun Choi[2], Sanghyun Hong[3], Dongeun Lee[4], Kookjin Lee[5], Noseong Park[6]**
[1]Yonsei University, [2]Seoul National University, [3]Oregon State University,
[4]Texas A&M University-Commerce, [5]Arizona State University, [6]KAIST
{jjsjjs0902, tuzi04}@yonsei.ac.kr, jonghyunchoi@snu.ac.kr
sanghyun.hong@oregonstate.edu, dongeun.lee@tamuc.edu,
kookjin.lee@asu.edu, noseong@kaist.ac.kr

## Abstract

A standard practice in developing image recognition models is to train a model on a specific image resolution and then deploy it. However, in real-world inference, models often encounter images different from the training sets in resolution and/or subject to natural variations such as weather changes, noise types and compression artifacts. While traditional solutions involve training multiple models for different resolutions or input variations, these methods are computationally expensive and thus do not scale in practice. To this end, we propose a novel neural network model, parallel-structured and all-component Fourier neural operator (PAC-FNO), that addresses the problem. Unlike conventional feed-forward neural networks, PAC-FNO operates in the frequency domain, allowing it to handle images of varying resolutions within a single model. We also propose a two-stage algorithm for training PAC-FNO with a minimal modification to the original, downstream model. Moreover, the proposed PAC-FNO is ready to work with existing image recognition models. Extensively evaluating methods with seven image recognition benchmarks, we show that the proposed PAC-FNO improves the performance of existing baseline models on images with various resolutions by up to 77.1% and various types of natural variations in the images at inference.

## 1 Introduction

Deep neural networks have enabled many breakthroughs in visual recognition (Simonyan & Zisserman, 2014; He et al., 2016; Szegedy et al., 2016; Krizhevsky et al., 2017; Dosovitskiy et al., 2020; Liu et al., 2022). A common practice of developing these models is to learn a model on training images with a *fixed* input resolution and then deploy the model to many applications.

In practice, when these models are deployed to real world, they are likely to face low-quality inputs at inference, *e.g.*, images with resolutions different from the training data and/or those with natural input variations such as weather changes, noise types, and compression artifacts. The use of such low-quality inputs significantly degrades the performance of visual recognition models. For example, Figure 1 shows that the ConvNeXt models (Liu et al., 2022) trained on ImageNet-1k (Russakovsky et al., 2015) suffer from (top-1) accuracy degradation when their inputs are of low-quality.

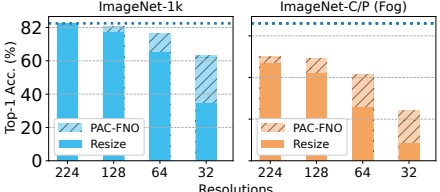

Figure 1: **Performance improvements by PAC-FNO in ImageNet-1k and ImageNet-C/P (Fog).** PAC-FNO improves the top-1 accuracy of ConvNeXt models when low-quality inputs are used, compared to the 'resize' baselines which is *resize-and-feed* using interpolation. ImageNet-C/P (Fog) is a dataset on which the 'fog' variation is applied to the inputs.

A naïve approach for using low-resolution images is to *resize-and-feed* them using interpolation methods. However, the figure shows that even when we use the best-performing interpolation method, the model's performance degrades significantly (*i.e.*, the images with a 32×32 resolution resized to 224×224 decreases the top-1 accuracy by 58%). Similarly, we observe the performance decrease in the ConvNeXt models if the images in weather changes such as fog. It becomes much worse when multiple variations are combined; for example, resizing 32×32 inputs containing the foggy weather condition leads to a decrease in the top-1 accuracy of the ConvNeXt models by 87%.

Here, we propose a novel architecture that can significantly alleviate the performance degradation of visual recognition models under low-quality inputs. (1) Unlike the prior work (Geirhos et al., 2018; Wang et al., 2020; Zhu et al., 2020; Yan et al., 2022a;b), our method addresses *both* the low-resolutions and input data variations at the same time. (2) Compared to existing methods (Haris et al., 2018; Anwar & Barnes, 2020) that require training multiple super-resolution models to accommodate different input resolutions (and/or input data variations), we only need a *single* model to support these input quality degradations.

To address both issues, we develop parallel-structured and all-component Fourier neural operator (PAC-FNO). By operating in the frequency domain, PAC-FNO is able to learn the semantics of images in various resolutions and/or natural variations for challenging image recognition with a single model. The proposed PAC-FNO is ready to work with deep neural visual recognition models to improve their performance.

We perform a comprehensive evaluation of PAC-FNO with seven image recognition benchmarks under various input quality degradation. In the evaluation, PAC-FNO demonstrates superior performance over other baselines in multiple variations such as weather changes at low resolution, with a performance increase of at least 11% at a 32×32 resolution.

**Contributions**. We summarize our contributions as follows:

- We propose PAC-FNO, a novel neural network architecture that operates in the frequency domain and offers resilience to image quality degradation, such as resolution changes and/or natural variations, a visual recognition model typically faces in real-world settings.
- Our design choices of PAC-FNO can offer such resilience with a *single* model, able to attach to a downstream visual recognition model. Not only does this handle multiple input variations at once, but the approach also minimizes the changes in the downstream model during fine-tuning.
- We perform an extensive evaluation of PAC-FNO with seven benchmarking tasks. We show that existing image recognition models, fine-tuned with PAC-FNO, can handle any input resolution degradation and are resilient to input variations that occur in real-world deployment settings.

## 2 PRELIMINARIES AND RELATED WORK

**Visual recognition with low-quality images.** Although low-quality images can be characterized by both low resolution and quality degradation, previous studies have only considered: i) resolution degradation or ii) quality degradation due to noise and weather changes, to name a few. For low-resolution images, they design a specific network for low-resolution classification (Zhu et al., 2020; Yan et al., 2022a;b) or attach an additional model that can handle low-resolution images, such as super-resolution (SR) models, in front of the pre-trained classification models (Cai et al., 2019). For quality degradation, an approach to solving this problem is to use images for training, which makes the model robust to image variations (Geirhos et al., 2018; Wang et al., 2020). Although they are successful in handling low resolution and input variations, respectively, to our knowledge, there is no method that considers the co-existence of resolution degradation and input variation

**Fourier neural operators (FNOs).** The FNOs achieve remarkable performance in solving PDEs with small computational costs and have been adopted by many other applications (Guibas et al., 2021; Li et al., 2020a;b; Rahman et al., 2022). First of all, FNOs are mathematically defined under the infinite dimensional continuous space regime and for this reason, they can process various resolutions of the continuous space without model changes. However, since the infinite dimensional continuous function of FNOs cannot be implemented in modern computers, the real implementation of FNOs follows the following quantized version:

$$\mathbf{h}_\ell = \sigma\big(\mathcal{F}^{-1}(\mathrm{R}_\theta \odot \xi(\mathcal{F}(\mathbf{h}_{\ell-1}))) + \mathbf{h}_{\ell-1}\mathbf{W}\big), \tag{1}$$

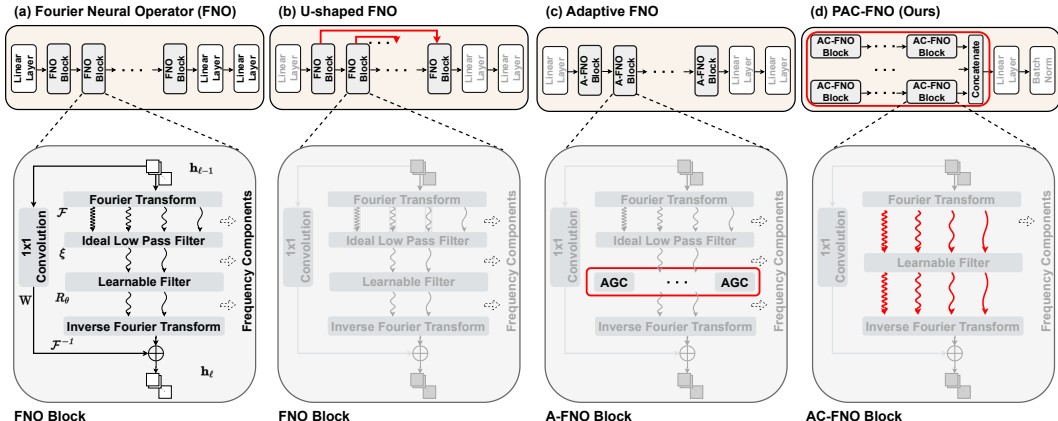

Figure 2: **Comparison of parallel-structured and all-component FNO (PAC-FNO) with existing FNOs.** (a) illustrates the vanilla FNOs. Each FNO block contains an ideal low-pass filter, a learnable filter ($R_\theta$) operating in the frequency domain, and a $1 \times 1$ convolutional operator ($W$). FNOs can have a series of these blocks. (b) shows a U-shaped FNO (UNO), connecting the FNO blocks in U-shape. (c) depicts Adaptive FNO, which replaces the learnable filter in the FNO block with adaptive global convolution (AGC). Our PAC-FNO shown in (d) uses all frequency components by removing the low-pass filter of the FNO block and runs forward with the AC-FNO block in parallel. (architecture advances are shown from (a) to (d) in red).

where $\mathbf{h}_\ell \in \mathbb{R}^{C_{in} \times H \times W}$ is a hidden vector at $\ell$-th layer. The kernel operator $\mathcal{F}(\kappa)$ is a Fourier transform that transforms the hidden vector into the frequency domain. $\xi$ is an ideal low-pass filter that removes signal components whose frequencies are greater than a threshold from the hidden vector. $\odot$ denotes Hadamard product with a learnable filter $R_\theta$ in the frequency domain. The kernel operator $\mathcal{F}^{-1}(\kappa)$ is an inverse Fourier transform. $W$ is a $1 \times 1$ convolutional operation. $\sigma$ is a non-linear activation function. For efficiency, FNOs restrict the size of the parameter $R_\theta$ by using an ideal low-pass filter. Then, $R_\theta \in \mathbb{C}^{C_{in} \times C_{out} \times H_r \times W_r}$, where $C_{in}(C_{out})$ is the channel size of the input (output) hidden vector and $H_r(W_r)$ means the height (width) after the ideal low-pass filter in the frequency domain (cf. Figure 2 (a)).

**FNOs for visual recognition.** FNOs' resolution invariance is attractive for visual recognition. Recently, applying FNOs to visual recognition has received attention (Guibas et al., 2021; Rahman et al., 2022; Viswanath et al., 2022; Pan et al., 2022; Wu et al., 2023), such as replacing the Vision Transformer's token mixer or detecting embers during wildfire, etc. Among them, U-shaped FNOs (Rahman et al., 2022) change the block configuration of FNOs, which is serial in FNOs, to an encoder and decoder structure using skip connection (cf. Figure 2 (b)).

In addition, adaptive FNO block (Guibas et al., 2021) replaces a token mixer of the Vision Transformer's self-attention (Dosovitskiy et al., 2020) by mixing tokens in the frequency domain. They improved memory and computational efficiency by replacing the learnable filter in the FNO block, which is not suitable for high-resolution input, with small-sized filters interpreted by the adaptive global convolution (AGC) operator. They parallelized AGCs to increase computational efficiency. Here, for a fair comparison, the A-FNO block is set to the same structure as the FNO (cf. Figure 2 (c)). As briefly reviewed here, FNOs are used as one part of the visual recognition model (*e.g.*, token mixing), and their use in visual recognition is still in the early stage of research.

## 3 PAC-FNO: PARALLEL-STRUCTURED AND ALL-COMPONENT FOURIER NEURAL OPERATORS

Despite the success of FNOs in visual recognition (Rahman et al., 2022; Viswanath et al., 2022; Wu et al., 2023), there are significant challenges in applying it to image classification. Wang et al. (2020) shows that there is a trade-off between model performance and generalization depending on image frequency components captured by the neural network in image classification. If the model

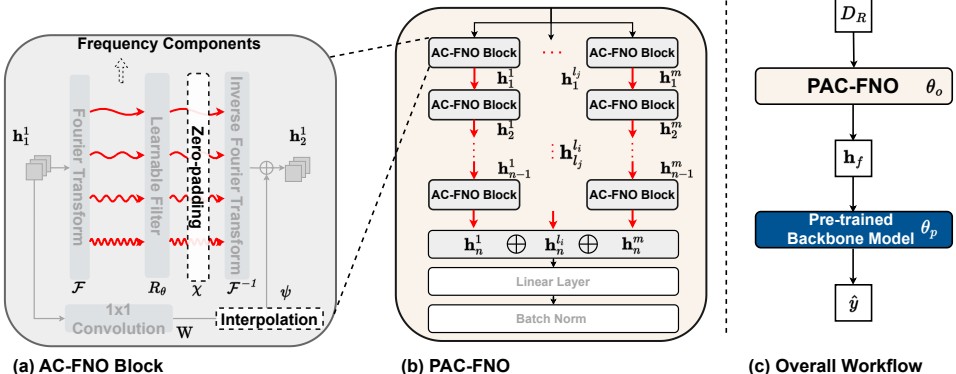

**(a) AC-FNO Block**      **(b) PAC-FNO**      **(c) Overall Workflow**

Figure 3: **Parallel-structured and all-component Fourier neural operator (PAC-FNO) architecture in detail.** (a) AC-FNO blocks use all frequency components and rely on Zero-padding and Interpolation to construct the images for the target resolution. (b) Contrary to previous FNOs, PAC-FNO consists of multiple AC-FNO blocks in a parallel manner. $\mathbf{h}_{l_j}^{l_i}$ is $(l_i, l_j)$-th hidden vector $(l_i \in \{0, \ldots, m\}, l_j \in \{0, \ldots, n\})$. In (c), $D_R$ is a set of image datasets with different resolutions, $h_f$ is a hidden vector processed from PAC-FNO, and $\hat{y}$ is a predicted class (see §3.1 and 3.2 for more details about PAC-FNO).

captures low-frequency components, the accuracy of the model increases, whereas if it captures high-frequency components, generalization improves.

However, to reduce the complexity of post-processing in FNO, an ideal low-pass filter is used after the Fourier transform (Li et al., 2020a), which removes high-frequency components that represent detail in the image. We argue that the low pass filter removes useful information for generalization of the model and harms the image classification accuracy. Thus, we propose a FNO block without the low pass filter, and call it 'all component' FNO (AC-FNO). We then propose to stack the AC-FNO blocks in a parallel structure to increase the capacity of encoding spatial information of images, call the whole structure as 'parallel' AC-FNO or PAC-FNO. We depict its architecture in Figure 3.

## 3.1 AC-FNO BLOCK

When solving PDEs describing fluid dynamics with FNOs, the input signal is assumed periodic in the frequency domain. For efficiency, non-significant parts are removed using a low-pass filter (in other words, high-frequency information is removed). However, in the case of images, high-frequency information sometimes plays an important role in image classification, especially when detailed information is required (type of bird, type of car, etc.). To this end, we propose an AC-FNO block without any band pass filters. Formally, our AC-FNO block is written as:

$$\mathbf{h}_\ell = \sigma\big(\mathcal{F}^{-1}(\chi(\mathrm{R}_\theta \odot \mathcal{F}(\mathbf{h}_{\ell-1}))) + \psi(\mathbf{h}_{\ell-1}\mathbf{W})\big), \tag{2}$$

where $\mathrm{R}_\theta \in \mathbb{C}^{3 \times 3 \times \mathrm{H_r} \times \mathrm{W_r}}$ denotes a learnable filter that maintains 3 channels of the image where $H_r$ and $W_r$ denotes the height and the width of the image (resolution) in the frequency domain, $\chi$ is an optional zero-padding, and $\psi$ is an optional interpolation to scale up the processed outcomes for low-resolution inputs only — the sizes of the processed outcomes are scaled up to the target resolution for which the backbone model was trained. Therefore, the AC-FNO block allows the processing of images in various resolutions while maintaining all frequency components (cf. Figure 3 (a)).

## 3.2 PARALLEL CONFIGURATION OF AC-FNO BLOCKS

We now propose to configure AC-FNO blocks in a parallel structure to increase the capacity to learn various types of input variations. Previous FNO models have a serial structure, and the first layer of the model, which is directly related to data, consists of only one block. It has too small a capacity to consider all components of the data. Therefore, we increase the capacity of the first layer with the parallel configuration of AC-FNO blocks, allowing it to utilize all frequencies of the image, including high-frequency and low-frequency components.

There are $n \times m$ AC-FNO blocks, where $n$ is the number of stages and $m$ is the number of parallel AC-FNO blocks in a stage. We then concatenate all the $m$ processed outcomes with a linear layer, followed by a batch normalization. The overall flow of the AC-FNO block can be written as follows:

$$\mathbf{h}_{\ell_j}^{\ell_i} = \text{AC-FNO-Block}_{\ell_j}^{\ell_i}(\mathbf{h}_{\ell_j-1}^{\ell_i}), \forall \ell_i, \ell_j, \tag{3}$$

$$\mathbf{h}_f = \text{BN}(\text{Linear}(\mathbf{h}_n^1, \mathbf{h}_n^2, ..., \mathbf{h}_n^m)), \mathbf{h}_n^{\ell_i} \in \mathbb{R}^{3 \times H_t \times W_t}, \tag{4}$$

where $\text{AC-FNO-Block}_{\ell_j}^{\ell_i}$ is the $(\ell_i, \ell_j)$-th AC-FNO block in Eq. 2. $\mathbf{h}_n^{\ell_i} \in \mathbb{R}^{3 \times H_t \times W_t}$ is the last stage of the hidden vectors where $H_t$ ($W_t$) is the height (width) of the target resolution, Linear denotes the linear layer that transforms from $3 \times m$ channel to 3 channel, and BN denotes a batch normalization layer. $\mathbf{h}_f$ is an input to the following backbone model. See §4.3 for the impact of PAC-FNO configurations on input variation.

## 3.3 Two-stage Learning Algorithm

To leverage the pre-trained model in training a PAC-FNO model, we propose a two-stage training algorithm for stable training. Specifically, we jointly train the PAC-FNO and the pre-trained backbone model together using only the target resolution dataset ($D_{target}$) for which the backbone model was trained. Since the pre-trained backbone model may not be able to fully understand the hidden space created by the PAC-FNO, we train them in a two-stage algorithm.

In the first phase, if the PAC-FNO and backbone model are well harmonized[1] for target resolution, the second phase begins. In the second phase, we fine-tune the well-harmonized model with images in low resolutions to generate a unified hidden space for all resolutions. If we use three low-resolution datasets with different resolutions for training, such as 32, 64, and 128, then $R$ in $D_R$ has a set of $\{target, low_1, low_2, low_3\}$. In AC-FNO blocks, interpolation and zero-padding operations are optionally performed on low-resolution images to match the size of the processing result. Then, $\mathbf{h}_f$, the output of PAC-FNO, is a hidden vector that represents commonalities in images of various resolutions and is used as input to a pre-trained backbone model to predict the class ($\hat{y}$).

In both stages, we use the cross-entropy loss for training. See §4.3 for the effectiveness of our proposed two-phase training algorithm. We describe the two-stage training algorithm in the Appendix (Algorithm 1) for the sake of space.

In addition, the computational cost of PAC-FNO is small since the number of parameters in the PAC-FNO is 1–13% of that of the backbone model. The parameters of PAC-FNO for various backbone models are reported in the Appendix §F.2.

## 4 Evaluation

For empirical validations, we evaluate PAC-FNOs and other state of the arts for low-quality visual recognition. After we train them with a pre-trained backbone model, we evaluate the model in low-resolution and input variations tasks at the same time. We then investigate the benefit of proposed components by ablation studies, and analyze the PAC-FNO further by sensitivity studies. Detailed setup is found in Appendix §F.1.

**Datasets.** We use seven image recognition benchmark datasets to evaluate PAC-FNO. For low-resolution tasks, six image recognition benchmark datasets are used to evaluate PAC-FNO: ImageNet-1k (Russakovsky et al., 2015), Stanford Cars (Krause et al., 2013), Oxford-IIIT Pets (Parkhi et al., 2012), Flowers (Nilsback & Zisserman, 2008), FGVC Aircraft (Maji et al., 2013), and Food-101 (Bossard et al., 2014). For input variation task, we use ImageNet-C/P (Hendrycks & Dietterich, 2019) which is an image dataset containing 19 common corruptions and perturbations.

**Backbone architectures.** We use four image classification models pre-trained on ImageNet-1k: two convolutional neural networks, ResNet-18 (He et al., 2016) and Inception-V3 (Szegedy et al., 2016), Vision Transformer (ViT) (Dosovitskiy et al., 2020) for a different architecture, and ConvNeXt-Tiny (Liu et al., 2022).

---

[1]Well harmonized means that the performance of the model combining PAC-FNO and the pre-trained backbone model is similar to that of a pre-trained model at the target resolution.

Table 1: **Performance of PAC-FNO on the low-resolution tasks using ImageNet-1k.** We report top-1 accuracy on low-resolution images generated from ImageNet-1k.

| Model | Method | Resolution | | | | | | | |
|---|---|---|---|---|---|---|---|---|---|
| | | 28 | 32 | 56 | 64 | 112 | 128 | **224** | **299** |
| ResNet-18 | Resize | 16.7 | 22.1 | 45.7 | 50.5 | 63.7 | 65.5 | 69.8 | - |
| | Fine-tune | 1.01 | 2.15 | 10.2 | 16.3 | 34.5 | 50.6 | 65.0 | - |
| | DRLN | 0.22 | - | 17.1 | - | 62.8 | - | 69.8 | - |
| | DRPN | 31.6 | - | 55.6 | - | 67.5 | - | 69.8 | - |
| | FNO | 40.2 | 45.2 | 59.1 | 61.5 | 67.6 | 68.5 | 70.1 | - |
| | UNO | 39.9 | 45.3 | 58.9 | 61.3 | 67.1 | 68.1 | 69.4 | - |
| | A-FNO | **43.3** | **49.5** | 60.2 | 62.5 | 66.9 | 67.5 | 68.5 | - |
| | PAC-FNO | 42.7 | 47.7 | **60.5** | **62.8** | **68.3** | **69.1** | **70.2** | - |
| Inception-V3 | Resize | 16.7 | 22.0 | 48.6 | 53.9 | 69.5 | 71.8 | - | 77.3 |
| | Fine-tune | 39.6 | 47.2 | 63.7 | 69.8 | 72.9 | 73.3 | - | 77.5 |
| | FNO | 48.9 | 54.0 | 68.5 | 70.2 | 74.9 | 76.5 | - | **78.4** |
| | UNO | 42.2 | 48.1 | 65.4 | 68.0 | 74.7 | 75.5 | - | 77.3 |
| | A-FNO | 33.4 | 39.6 | 59.6 | 62.7 | 71.0 | 72.1 | - | 74.9 |
| | PAC-FNO | **49.3** | **54.9** | **68.8** | **70.7** | **76.1** | **76.9** | - | **78.4** |

| Model | Method | Resolution | | | | | | | |
|---|---|---|---|---|---|---|---|---|---|
| | | 28 | 32 | 56 | 64 | 112 | 128 | **224** | **299** |
| ViT-B16 | Resize | 41.6 | 47.4 | 66.0 | 68.9 | 76.8 | 78.1 | **81.1** | - |
| | Fine-tune | 39.8 | 48.5 | 66.6 | 69.6 | 77.3 | 78.4 | 81.1 | - |
| | DRLN | 3.75 | - | 42.0 | - | 77.0 | - | 81.1 | - |
| | DRPN | 52.2 | - | **73.0** | - | **80.0** | - | 81.1 | - |
| | FNO | 39.1 | 46.6 | 66.2 | 69.2 | 77.1 | 78.0 | 79.7 | - |
| | UNO | 43.7 | 50.6 | 67.9 | 70.3 | 77.9 | 78.8 | 80.2 | - |
| | A-FNO | **54.9** | **59.5** | 72.3 | **74.1** | 78.3 | **78.9** | 79.5 | - |
| | PAC-FNO | 46.6 | 52.1 | 69.0 | 71.6 | 77.7 | 78.6 | 79.6 | - |
| ConvNeXt-Tiny | Resize | 27.5 | 34.5 | 60.7 | 64.9 | 75.0 | 77.3 | **82.5** | - |
| | Fine-tune | 40.2 | 62.3 | 65.8 | 76.0 | 76.4 | **80.7** | 81.8 | - |
| | DRLN | 0.30 | - | 25.1 | - | 71.8 | - | 82.5 | - |
| | DRPN | 40.7 | - | 68.2 | - | 79.4 | - | 82.5 | - |
| | FNO | 42.8 | 48.9 | 67.4 | 69.8 | 76.5 | 77.4 | 79.2 | - |
| | UNO | 58.6 | 62.9 | 73.7 | 75.5 | 79.1 | 79.7 | 80.6 | - |
| | A-FNO | 48.3 | 53.5 | 68.1 | 70.5 | 75.9 | 76.6 | 78.1 | - |
| | PAC-FNO | **58.9** | **63.2** | **74.5** | **76.2** | **80.2** | 80.7 | 81.5 | - |

**Baselines.** We use three baseline types: i) resizing, ii) super-resolution (SR) models, and iii) Fourier neural operator models. Resizing includes Resize and Fine-tune methods that do not require additional networks. Resize is a method that directly feeds the resized images to a pre-trained classification model using interpolation, while Fine-tune is a method of fine-tuning the pre-trained classification model with the resized images. SR models include DRLN (Anwar & Barnes, 2020) and DRPN (Haris et al., 2018), and these two models are representative models capable of up to 8 times super-resolution. With SR models, low-resolution images are upscaled to the target resolution and fed into a pre-trained model. Fourier neural operator models include vanilla FNO (Li et al., 2020a), UNO (Rahman et al., 2022), and A-FNO (Guibas et al., 2021). These models are most similar to our PAC-FNO and are trained by our proposed training method for fair comparison.

**Metrics.** We use two metrics for the evaluation: top-1 accuracy and relative accuracy. Relative accuracy is defined as the ratio of a model's accuracy in low quality to that in the original resolution. This metric enables us to compare the effectiveness of methods in the same resolution across different datasets and models.

**Image degradation types.** We generate low-resolution images by resizing and cropping the original images. We consider resolutions in $\{28, 32, 56, 64, 112, 128, 224, 299\}$, which are frequently used in image classification. For training we use $\{224\}$ or $\{299\}$ as target resolution depending on the backbone model along with $\{32, 64, 128\}$ low resolution. During inference, seven resolutions are used, including three datastes not used for training. Super-resolution models can only be upscaled by 2, 4, or 8 times, so they can be inferred only at resolutions of 28, 56, and 112.

## 4.1 Low-resolution image recognition

Table 1 shows the top-1 accuracy results of ImageNet-1k. In most cases, PAC-FNO shows good performance. In particular, PAC-FNO shows the best performance not only at low resolution but also at the target resolution of 299 in Inception-V3. We also find that in ConvNeXT-Tiny, PAC-FNO shows the best accuracy at all low resolutions, even if the result at the target resolution is lower than the Resize method (81.5 vs. 82.5). This suggests that PAC-FNO shows good performance in terms of relative accuracy which we proposed. The results of relative accuracy are in the Appendix §F.5.

For fine-grained image datasets, ConvNeXt-Tiny pre-trained with ImageNet-1k is fine-tuned for each dataset with a target resolution. After that, the same algorithm used in ImageNet-1k is applied to the fine-tuned ConvNeXt-Tiny model. Table 2 shows the experimental results for four fine-grained datasets. The proposed method, PAC-FNO, shows good performance at low resolutions by a large margin for all datasets.

In particular, among the Fourier neural operator models, only the PAC-FNO shows better performance than Fine-tune method. This represents the effectiveness of the ideal low-pass filter in the FNO block, which removes detailed image signals that play an important role in classification in the fine-grained dataset. In Food-101, the target resolution accuracy is 2% lower than Fine-tune method,

Table 2: **Performance of PAC-FNO on the low-resolution tasks using fine-grained datasets.** We report top-1 accuracy for the ConvNext-Tiny model on low-resolution images generated from fine-grained datasets.

| Dataset | Method | Resolution | | | | | | |
|---|---|---|---|---|---|---|---|---|
| | | 28 | 32 | 56 | 64 | 112 | 128 | **224** |
| Oxford-IIIT Pets | Resize | 29.4 | 36.4 | 70.1 | 77.0 | 89.9 | 91.2 | 93.7 |
| | Fine-tune | 32.6 | 41.1 | 73.2 | 79.3 | 91.0 | **91.5** | 93.8 |
| | DRLN | 3.37 | - | 36.8 | - | 88.1 | - | 93.7 |
| | DRPN | 41.5 | - | 84.2 | - | **92.6** | - | 93.7 |
| | FNO | 19.1 | 54.0 | 60.6 | 70.2 | 86.7 | 90.4 | 91.4 |
| | UNO | 11.1 | 15.7 | 43.2 | 50.5 | 80.1 | 83.8 | 90.3 |
| | A-FNO | 27.0 | 33.3 | 63.5 | 70.2 | 85.2 | 86.9 | 89.1 |
| | PAC-FNO | **73.4** | **77.3** | **86.0** | **87.6** | 90.5 | 91.1 | 91.7 |
| Flowers | Resize | 39.5 | 47.6 | 75.3 | 80.3 | 92.1 | 93.7 | 95.9 |
| | Fine-tune | 44.3 | 51.1 | 78.8 | 81.9 | 92.5 | 94.0 | 95.9 |
| | DRLN | 9.92 | - | 53.4 | - | 91.3 | - | 95.9 |
| | DRPN | 64.5 | - | 89.0 | - | **95.0** | - | 95.9 |
| | FNO | 25.3 | 32.8 | 65.9 | 73.0 | 91.7 | 93.7 | **96.6** |
| | UNO | 23.2 | 30.2 | 64.5 | 71.7 | 90.8 | 92.6 | 96.1 |
| | A-FNO | 26.8 | 33.6 | 63.7 | 70.0 | 86.2 | 88.8 | 91.9 |
| | PAC-FNO | **74.1** | **78.0** | **87.6** | **89.3** | 93.6 | **94.2** | 94.4 |
| FGVC Aircraft | Resize | 2.46 | 2.76 | 27.3 | 41.8 | 71.0 | 74.3 | 79.7 |
| | Fine-tune | 8.5 | 16.2 | 40.2 | 58.3 | 77.2 | 76.0 | 79.8 |
| | DRLN | 1.14 | - | 13.1 | - | 68.9 | - | 79.7 |
| | DRPN | 18.4 | - | 59.6 | - | **77.0** | - | 79.7 |
| | FNO | 26.3 | 33.2 | 59.4 | 64.0 | 74.6 | 75.7 | 80.7 |
| | UNO | 1.65 | 1.70 | 9.17 | 22.7 | 73.0 | 76.4 | **81.6** |
| | A-FNO | 6.78 | 11.5 | 50.8 | 59.2 | 75.1 | 77.0 | 80.8 |
| | PAC-FNO | **37.7** | **45.1** | **67.0** | **69.4** | **77.0** | **78.6** | 81.1 |
| Food-101 | Resize | 40.1 | 48.1 | 76.2 | 80.2 | 88.1 | 89.2 | 91.1 |
| | Fine-tune | 50.2 | 54.1 | 80.3 | 82.0 | 88.3 | 89.6 | **91.2** |
| | DRLN | 7.06 | - | 39.1 | - | 86.8 | - | 91.1 |
| | DRPN | 54.8 | - | 81.6 | - | 59.5 | - | 91.1 |
| | FNO | 43.6 | 51.3 | 77.9 | 81.0 | 88.2 | 89.0 | 90.8 |
| | UNO | 50.5 | 56.7 | 75.5 | 78.3 | 85.2 | 86.0 | 87.9 |
| | A-FNO | 46.0 | 51.7 | 72.2 | 75.8 | 83.8 | 84.8 | 88.2 |
| | PAC-FNO | **74.9** | **77.8** | **85.5** | **86.9** | **89.2** | **89.7** | 90.4 |

Table 3: **Performance of PAC-FNO on the input variation tasks.** We report top-1 accuracy for the ConvNext-Tiny model on four natural input corruptions, chosen from ImageNet-C/P (Hendrycks & Dietterich, 2019).

| Variation | Method | Resolution | | | | | | |
|---|---|---|---|---|---|---|---|---|
| | | 24 | 32 | 54 | 64 | 112 | 128 | 224 |
| Fog | Resize | 8.21 | 10.8 | 27.2 | 31.9 | 48.5 | 52.3 | 58.4 |
| | Fine-tune | 23.2 | 28.2 | 47.5 | 51.4 | **61.0** | **62.2** | **63.0** |
| | DRLN | 0.16 | - | 6.54 | - | 33.5 | - | 58.4 |
| | DRPN | 0.67 | - | 0.99 | - | 1.32 | - | 58.4 |
| | FNO | 14.4 | 18.6 | 38.9 | 43.5 | 56.6 | 58.4 | 60.3 |
| | UNO | 24.6 | 29.7 | 46.8 | 49.5 | 59.2 | 60.4 | 59.0 |
| | A-FNO | 11.0 | 14.9 | 33.4 | 38.3 | 51.9 | 53.6 | 53.3 |
| | PAC-FNO | **25.4** | **30.4** | **48.2** | **51.7** | 60.1 | 61.4 | 62.8 |
| Brightness | Resize | 22.0 | 28.2 | 52.7 | 56.8 | 67.8 | 69.9 | 73.5 |
| | Fine-tune | 42.8 | 48.9 | 65.8 | 68.1 | **73.5** | **74.4** | **75.7** |
| | DRLN | 6.64 | - | 35.71 | - | 56.1 | - | 58.4 |
| | DRPN | 3.09 | - | 1.08 | - | 8.01 | - | 58.4 |
| | FNO | 34.9 | 40.7 | 59.4 | 62.6 | 70.0 | 71.3 | 73.1 |
| | UNO | 50.8 | 55.6 | 66.7 | 67.9 | 71.4 | 72.1 | 72.8 |
| | A-FNO | 37.6 | 43.2 | 59.4 | 62.5 | 68.8 | 69.8 | 70.0 |
| | PAC-FNO | **51.9** | **56.5** | **68.2** | **69.8** | 73.4 | 74.1 | 74.7 |
| Spatter | Resize | 18.5 | 22.9 | 40.7 | 44.1 | 56.3 | 58.9 | 62.6 |
| | Fine-tune | 35.6 | 39.8 | 52.5 | 54.8 | **61.8** | **63.0** | **64.6** |
| | DRLN | 1.85 | - | 17.5 | - | 37.3 | - | 62.6 |
| | DRPN | 0.83 | - | 0.36 | - | 0.34 | - | 62.6 |
| | FNO | 28.5 | 31.9 | 44.9 | 47.8 | 57.5 | 59.4 | 62.4 |
| | UNO | 40.2 | 42.9 | 50.5 | 52.2 | 57.7 | 58.8 | 59.8 |
| | A-FNO | 32.0 | 35.0 | 45.8 | 48.4 | 55.9 | 57.4 | 59.2 |
| | PAC-FNO | **43.2** | **46.0** | **54.5** | **56.4** | 61.3 | 62.3 | 64.0 |
| Saturate | Resize | 17.0 | 22.6 | 47.5 | 52.2 | 64.7 | 67.1 | 73.5 |
| | Fine-tune | 35.6 | 41.7 | 61.4 | 64.5 | 71.4 | 72.5 | 74.3 |
| | DRLN | 1.26 | - | 19.4 | - | 44.5 | - | 73.5 |
| | DRPN | 0.78 | - | 0.56 | - | 0.72 | - | 73.5 |
| | FNO | 30.3 | 36.0 | 55.7 | 59.0 | 67.3 | 68.7 | 70.5 |
| | UNO | 45.8 | 50.7 | 63.5 | 65.3 | 70.2 | 71.0 | 70.8 |
| | A-FNO | 37.9 | 43.3 | 59.3 | 62.1 | 68.2 | 69.0 | 69.3 |
| | PAC-FNO | **44.2** | **56.5** | **64.5** | **69.8** | **71.9** | **74.1** | **74.7** |

but PAC-FNO shows the highest accuracy in all other low resolutions. Compared to the Fine-tune method, accuracy increased by 48.3%, 43.8%, 6.48%, and 5.98%, at 28, 32, 56, and 64 resolutions, respectively. Results for the rest of the fine-grained datasets and best hyperparameter settings are in the Appendix §F.4 and §E. Additionally, we conducted a fine-grained classification experiment on Oxford-IIIT Pets with the ViT model (cf. Appendix §F.4).

## 4.2 RESILIENCE TO NATURAL INPUT VARIATIONS

We now test the resilience of input data variations in test-time, such as weather changes, that likely occurs in real-world deployment settings. We run experiments with ImageNet-C/P (Hendrycks & Dietterich, 2019). ImageNet-C/P has 19 noises that can be used to test resilience against image quality degradation. Table 3 summarizes the top-1 accuracy of PAC-FNO under four input variations—fog, brightness, spatter, and saturate.

PAC-FNO shows superior performance in most cases, especially at low resolution. Only the Fine-tune method shows slightly better performance than PAC-FNO at a target resolution of 224 in fog and brightness. However, the Fine-tune method in fog has a greater performance degradation than PAC-FNO at low resolution. In particular, at $32 \times 32$ resolution, there is a 7.6% performance difference (35.6% vs. 43.2%) between Fine-tune and PAC-FNO. Table 3 shows that the resolution invariant models show good performance at low resolutions and Resize and Fine-tune methods show good performance at relatively high resolutions. In the case of SR models, performance is completely reduced at low-resolution input variation. Our proposed PAC-FNO is more resilient to input changes regardless of resolution than other baseline methods. Results for the rest of the fifteen input

variations are in Appendix §F.6 for the sake of space. We also show in Appendix F.8 an analysis of how the parallel structure of PAC-FNO affects input variation.

## 4.3 Ablation Studies

We now investigate the effect of PAC-FNO's contribution. We conduct experiments for parallel configurations of PAC-FNO and a two-stage training algorithm. The results of the ablation study of PAC-FNO with low and high pass filters were reported in Appendix §F.7 for the sake of space.

**Configurations of PAC-FNO layer.** We investigate the benefit of our parallel structure to increase the capacity to learn various types of input variants by comparing the performance of the parallel and serial configurations using eight AC-FNO blocks so that total number of AC-FNO blocks are the same but in difference configuration.

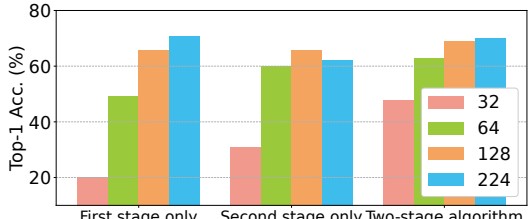

For the experiment, ConvNeXt-Tiny is used for the backbone network with ImageNet-1k and ImageNet-1k with fog degradation. Figure 4 shows that the parallel configuration increases performance in terms of accuracy compared to the serial configuration with the same number of AC-FNO blocks on both datasets at all resolutions. It implies that the parallel structure has a clear benefit to better encode the images.

Figure 4: **Benefit of parallel structure.** We show the top-1 performance of PAC-FNO models with different configurations. $n$ and $m$ are the number of blocks in series and parallel, respectively. Note that the total number of AC-FNO blocks is the same for both configurations. We use ImageNet-1k and ImageNet-C/P (Fog). Compared to serial configuration, our proposed parallel configuration shows less performance degradations at target resolution (39.1% vs. 22.9%)

In addition, Figure 4 shows the resilience to input variations depending on the PAC-FNO configurations. In target resolution (224), the performance decrease is reduced in parallel configuration compared to serial configuration such as 22.9% (81.5% → 62.8%) in parallel and 39.3% (79.1% → 48.0%) in serial configuration. As expected, parallel architectures seem to consider more frequency components than serial architectures.

**Two-stage Training Algorithm for PAC-FNO.** We also conducted an ablation study on our proposed two-stage training algorithm. We compare the two-stage training algorithm with i) only the first training stage with the target resolution and ii) only the second training stage with low-resolutions without the first training stage.

We show the result in Figure 5. As shown in the figure, our two-stage training algorithm has a clear benefit. 'First stage only', which does not use low-resolution images but only trains with target resolution, shows a performance decrease at low resolution. 'Second stage only', which is trained directly with low-resolutions and target resolution without a first stage, shows higher performance than 'First stage only' at low-resolutions, but it shows that performance does not increase as the resolution increases, but that performance is good at intermediate resolutions. Without our full training scheme, the model does not converge well and the accuracy in its target resolution is severely degraded.

Figure 5: **Benefit of two-stage training algorithm.** We show the top-1 accuracy of the ablation study of the two-stage algorithm. 'First stage only' refers to a model that was trained only with the first stage, and 'Second stage only' refers to a model that was trained only with the second stage, and 'Two-stage algorithm' refers to a model that was trained by our two-stage training algorithm. We use ResNet-18 in ImageNet-1k.

## 4.4 Sensitivity Studies

We then investigate the performance according to the PAC-FNO configuration and the number of trained resolutions.

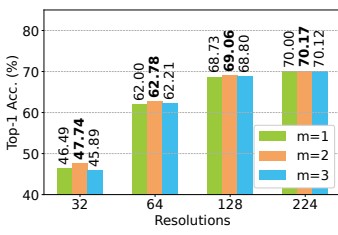 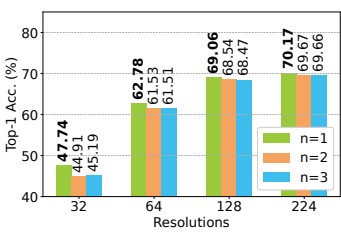

(a) ImageNet-1k ($n = 1$)   (b) ImageNet-1k ($m = 2$)

Figure 6: **Results for the number of stages $n$ and blocks $m$ in PAC-FNO.** We vary $n$ and $m$ from $\{1, 2, 4\}$, respectively, and show the model performance in low-resolution inputs.

Table 4: **Results for a number of seen resolutions during training.** We show the top-1 accuracy of ResNet-18 models according to the number of resolutions during training time. The image resolutions we used are on the left, and the inference resolutions we test are on the right. We use ImageNet-1k.

| Training Resolution | Inference Resolution | | | | | | | | Training Resolution | Inference Resolution | | | | | | | |
|---|---|---|---|---|---|---|---|---|---|---|---|---|---|---|---|---|---|
| | 32 | 48 | 64 | 96 | 128 | 160 | 192 | 224 | | 32 | 48 | 64 | 96 | 128 | 160 | 192 | 224 |
| $\{32, 64, 128, 224\}$ | 47.7 | 57.8 | 62.8 | 67.1 | 69.1 | 70.0 | 70.0 | 70.2 | $\{32, 224\}$ | 53.0 | 55.5 | 59.3 | 64.1 | 66.4 | 68.0 | 69.0 | 69.3 |

**Sensitivity study of PAC-FNO layer.**   We show the impact of changing the number of stages ($n$) and the number of AC-FNO blocks ($m$) constituting the PAC-FNO. While one of the $n$ and $m$ is controlled, the other parameter is manipulated, varying as 1, 2, and 4. For the experiment, ResNet-18 is used for the backbone network with ImageNet-1k.

As shown in Figure 6, the appropriate number of stages and AC-FNO blocks for ImageNet-1k is $n = 1$ and $m = 2$. $n$ and $m$ show differences depending on the backbone models and datasets, but $m \geq 2$ in all cases. In other words, we found that our proposed parallel configuration of AC-FNO blocks is effective in low-resolution image recognition. The best hyperparameters for PAC-FNO configuration are reported in the Appendix §B.

**Number of resolutions during training.**   We show the performance of PAC-FNO depending on the number of resolutions used for training. We use two models for comparison and report top-1 accuracy in Table 4. One is a model trained with resolutions frequently used in image classification, such as $\{32, 64, 128, 224\}$, and the other one is a model trained with extreme resolutions of $\{32, 224\}$. Both models show meaningful results even for resolutions that are not used for training.

However, the model trained with a wider number of resolutions shows better performance except for the results at $32 \times 32$ resolution. This shows that better performance can be achieved in real-world applications with various resolutions without a process to resize to the target resolution. In addition, this result shows the efficacy of our neural operator-based mechanism which has a strong point in utilizing images in real-world applications.

## 5   CONCLUSION

We proposed parallel-structured and all-component Fourier neural operators (PAC-FNOs) for visual recognition under low-quality images. To this end, we design i) an AC-FNO block without any band pass filters and ii) a parallel configuration of AC-FNO blocks with increased capacity to learn different types of input variations.

In addition, we also propose a two-stage training algorithm for the training of the stable PAC-FNO model. As a result, PAC-FNO provides two advantages over existing methods: (i) It can handle both low-resolution and input variations typically observed in low-quality images with a single model; (ii) One can attach PAC-FNO to any visual recognition model and fine-tune it. In the evaluation with four visual recognition models and seven datasets, we show that PAC-FNO achieves high accuracy for various resolutions and input variations. In particular, it shows the highest accuracy on low-quality inputs that combine multiple effects (*e.g.*, low-resolution images in foggy weather).

**Future Work.**   An interesting line of work is to apply PAC-FNO in complex real-world settings, *e.g.*, where multiple degradations of input can occur (*e.g.*, the low-resolution surveillance camera mixed with motion blur and fog).

## ETHICS STATEMENT

Our approach fine-tunes PAC-FNO and pre-trained image recognition models together, allowing a *single* model to handle all input resolution degradation and to be resilient to input changes that occur in real-world deployment settings. In the process of fine-tuning the carbon emissions of the machine for training and running, it is essential. However, this can significantly reduce carbon emissions compared to approaches that require training and running a resolution degradation model and an input variations model separately.

## REPRODUCIBILITY STATEMENT

For reproducibility, we attached the source codes in our supplementary materials. There are detailed descriptions for experimental environment settings, datasets, training processes, and evaluation processes in README.md. In the Appendix §E, we also list all the detailed hyperparameters.

### ACKNOWLEDGMENTS

This work was partly supported by the NRF grant (No.2022R1A2C4002300, 10%) and IITP grants (No.2020-0-01361 (20%, Yonsei AI), No.2021-0-01343 (10%, SNU AI), No.2022-0-00077 (10%), No.2022-0-00113 (40%), No.2021-0-02068 (10%, AI Innov. Hub)) funded by the Korea government (MSIT). Sanghyun is partially supported by the Google Faculty Research Award. Work is done when J. Choi and N. Park are with Yonsei University. J. Choi and N. Park are co-corresponding authors.

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

APPENDIX

## A  VARIOUS RESOLUTION IMAGES

There are no image datasets prepared for various resolutions. Therefore, creating images in various resolutions through an appropriate resizing method is important in our task. Chrabaszcz et al. (2017) showed that low-resolution images resized by some interpolations have the characteristics of the original image. Among various interpolation methods, the bilinear interpolation method retains the most characteristics of the original images. Therefore, we constructed low-resolution image datasets applying the same bilinear interpolation method to the original dataset.

After constructing the low-resolution datasets, we examined the performance decrease with various pre-trained models. To check the performance decrease, the low-resolution images were resized back to the target resolution (224) with various interpolation methods. Table 5 shows that the bicubic interpolation has the smallest performance decrease. As mentioned in Section 1, images resized by the bicubic interpolation were used as input to various pre-trained models.

Table 5: The top-1 accuracy of pre-trained ConvNeXt-Tiny in various resolutions when resized up with each interpolating method.

| Method | Acc. (%) | Resolution | | | |
|---|---|---|---|---|---|
| | | 32 | 64 | 128 | **224** |
| Nearest | Top-1 | 0.9 | 18.17 | 58.27 | **82.52** |
| Bilinear | Top-1 | 29.47 | 60.36 | 76.98 | **82.52** |
| Bicubic | Top-1 | **34.51** | **64.88** | **77.27** | **82.52** |
| Area | Top-1 | 0.9 | 34.92 | 74.42 | **82.52** |
| Nearest-exact | Top-1 | 0.9 | 18.27 | 58.27 | **82.52** |

Table 6: Classification accuracy drops when scaling up low-resolution images, *i.e.*, 32, 64, 128, to target resolution, *i.e.*, 224 or 299, using existing image interpolation methods. We report the best interpolation method's results.

| Model | Acc. (%) | Resolution | | | | | # Param. |
|---|---|---|---|---|---|---|---|
| | | 32 | 64 | 128 | **224** | **299** | |
| ResNet-18 | Top-1 | 22.09 | 50.53 | 65.52 | 69.76 | - | 11.69M |
| | Relative | 31.67 | 72.43 | 93.91 | 100 | - | |
| Inception-V3 | Top-1 | 22.05 | 53.92 | 71.77 | - | 77.29 | 27.16M |
| | Relative | 33.69 | 72.39 | 93.51 | - | 100 | |
| ViT-B16 | Top-1 | 47.39 | 68.91 | 78.13 | 81.07 | - | 86.57M |
| | Relative | 58.46 | 85.00 | 69.37 | 100 | - | |
| ConvNeXt-Tiny | Top-1 | 34.51 | 64.88 | 77.27 | 82.52 | - | 28.59M |
| | Relative | 42.02 | 79.00 | 94.08 | 100 | - | |

## B  DETAILED DESCRIPTION OF DATASETS

The following are detailed descriptions of the datasets used in our experiments. The number of classes, train images, and test images are organized in Table 7.

### B.1  IMAGENET-1K

ImageNet-1k (Russakovsky et al., 2015) is the most widely used dataset in image classification along with iNaturalist. In particular, the most widely used dataset, ImageNet Large Scale Visual Recognition Challenge 2012 (ILSVRC2012), contains 1000 categories of 1.2 million images. ImageNet is built with the WordNet hierarchy which means each category is explained by several words or phrases. Thus, its goal is to offer a set of qualified images in the same hierarchical words or phrases.

### B.2  FINE-GRAINED DATASETS

Fine-grained Datasets contain classes from a single category, such as cars, birds, flowers, aircraft, or food, and these are more difficult to classify where detailed elements must be considered.

**Stanford Cars** Stanford Cars (Krause et al., 2013) is a dataset containing car images. There are 196 classes with information on make, model, and year (*e.g.*, 2012 BMW M3 coupe).

**Oxford-IIIT Pets** Oxford-IIIT Pets (Parkhi et al., 2012) contains images of cats and dogs with 37 categories. The label consisted of species and breeds. This dataset also provides a bounding box for segmentation tasks.

**Flowers** Flowers (Nilsback & Zisserman, 2008) includes 102 categories of flowers that are mainly found in the United Kingdom. Even flowers belonging to the same categories have large variations in pose, size, and light.

**FGVC Aircraft** FGVC Aircraft (Maji et al., 2013) is used in the fine-grained recognition challenge 2013 (FGComp). Aircraft with various sizes, purposes, driving forces, and other features are classified by those fine-grained features. The 100 categories of aircraft include the make or name of the model series.

**Food-101** Food-101 (Bossard et al., 2014) contains 101 kinds of food images, which were originally provided to be used for RandomForest. The number of images is much larger compared to other fine-grained datasets. The images in this dataset are already rescaled and contain noises ranging from color intensity to wrong labels.

Table 7: The number of classes, train images and test images.

| Datasets | # of classes | # of train images | # of test images |
|---|---|---|---|
| ImageNet-1k | 1000 | 1,281,167 | 50,000 |
| Stanford Cars | 196 | 8,144 | 8,041 |
| Oxford-IIIT Pets | 37 | 3,680 | 3,669 |
| Flowers | 102 | 1,020 | 6,149 |
| FGVC Aircraft | 100 | 3,334 | 3,333 |
| Food-101 | 101 | 75,750 | 25,250 |

## C  TRAINING ALGORITHM

We describe a two-stage training algorithm in Algorithm. 1. We first stage for training with target resolution in the first while loop, followed by the second while loop for training with various resolutions.

---

**Algorithm 1:** 2-stage Training of PAC-FNO

**Input:** Target resolution data $D_{target}$, Number of low-resolutions $N$, Low resolution data $\{D_{low_1}, \cdots, D_{low_N}\}$, Class label $y$, Predicted class label $\hat{y}$, First phase iteration number $K_{first}$, Second phase iteration number $K_{second}$, PAC-FNO parameters $\boldsymbol{\theta}_o$, Pre-trained backbone model parameters $\boldsymbol{\theta}_p$, cross-entropy loss $CE$

Initialize $\boldsymbol{\theta}_o$;
$k \leftarrow 0$;
/* First training stage                                                    */
**while** $k < K_{first}$ **do**
  **for** *each mini-batch* $B \subseteq D_{target}$ **do**
    $\{\hat{\mathbf{y}}_i\}_{i=1}^{|B|} \leftarrow$ Backbone(PAC-FNO($B; \boldsymbol{\theta}_o$);$\boldsymbol{\theta}_p$);
    Train $\boldsymbol{\theta}_o$ and $\boldsymbol{\theta}_p$ with $CE(\{\hat{\mathbf{y}}_i\}_{i=1}^{|B|}, \{\mathbf{y}_i\}_{i=1}^{|B|})$;
  $k \leftarrow k + 1$;
**end**
$k \leftarrow 0$ ;
/* Second training stage                                                   */
**while** $k < K_{second}$ **do**
  **for** $i \leftarrow 1$ *to* $N$ **do**
    **for** *each mini-batch* $B \subseteq D_{low_i}$ **do**
      $\{\hat{\mathbf{y}}_i\}_{i=1}^{|B|} \leftarrow$ Backbone(PAC-FNO($B; \boldsymbol{\theta}_o$);$\boldsymbol{\theta}_p$);
      Train $\boldsymbol{\theta}_o$ with $CE(\{\hat{\mathbf{y}}_i\}_{i=1}^{|B|}, \{\mathbf{y}_i\}_{i=1}^{|B|})$;
  **for** *each mini-batch* $B \subseteq D_{target}$ **do**
    $\{\hat{\mathbf{y}}_i\}_{i=1}^{|B|} \leftarrow$ Backbone(PAC-FNO($B; \boldsymbol{\theta}_o$);$\boldsymbol{\theta}_p$);
    Train $\boldsymbol{\theta}_o$ with $CE(\{\hat{\mathbf{y}}_i\}_{i=1}^{|B|}, \{\mathbf{y}_i\}_{i=1}^{|B|})$;
  $k \leftarrow k + 1$;
**end**

---

## D  BACKBONE MODELS

We propose a plug-in module for multi-scale classification. Therefore, we applied various pre-trained classification models from CNN-based to ViT-based classification models. All classification models were trained with ImageNet-1k from scratch with the settings in Table 8. TORCHVISION provides all such pre-trained models.

**ResNet-18** Residual network (ResNet) is a model that applies the concept of residual connection to CNN architectures. ResNet-18 (He et al., 2016) consists of 18 convolutional blocks and 8 residual layers. It is the most fundamental model in image classification.

Table 8: Recipe for the pre-trained models provided in TORCHVISION.

| Model | ResNet-18 | ViT-B16 | ConvNeXt-Tiny |
|---|---|---|---|
| Epochs | 90 | 300 | 600 |
| Batch size | 32 | 512 | 128 |
| Optimizer | sgd | adamw | adamw |
| Learning rate (lr.) | 0.1 | 3e-3 | 1e-3 |
| Weight decay | 1e-4 | 0.3 | 0.05 |
| lr. scheduler | steplr | cosineannealinglr | cosineannealinglr |
| lr. warmup method | - | linear | linear |
| lr. warmup epochs | - | 30 | 5 |
| lr. warmup decay | - | 0.033 | 0.01 |
| Amp | - | ○ | - |
| Random erase | - | - | 0.1 |
| Label smoothing | - | 0.11 | 0.1 |
| Mixup alpha | - | 0.2 | 0.2 |
| Auto augment | - | ra | ta wide |
| Clip grad norm | - | 1 | - |
| Ra sampler | - | ○ | ○ |
| Cutmix alpha | - | 1.0 | 1.0 |
| Model-ema | - | ○ | ○ |
| Norm weight decay | - | - | 0.0 |
| Train crop size | 224 | 224 | 176 |
| Test resize size | 256 | 256 | 232 |
| Test crop size | 224 | 224 | 224 |
| Ra reps | - | 3 | 4 |

**Inception-V3** Inception networks (Szegedy et al., 2016) are one of the most popular classification models, which stacks deep convolutional layers in an efficient way. They proposed techniques such as the concatenation of convolutional layers with different kernel sizes, kernel decomposition, and backpropagation with an auxiliary classifier for efficient training. We used the pre-trained Inception-V3 model provided by TORCHVISION, and there is no training recipe for training from scratch.

**ViT-B16** Vision Transformer (Dosovitskiy et al., 2020) (ViT) is a model that uses transformers based on a self-attention architecture in the image domain. It divides an image into patches and calculates all patch-by-patch relationships through self-attention. Because of these calculations, much memory and training time are required. Among those ViT-based models, ViT-B16 has the smallest model size and divides an image into 16 patches.

**ConvNeXt-Tiny** ConvNeXt (Liu et al., 2022) is one of the most recent models with good performance using only convolutional networks. It shows better performance with fewer parameters compared to ViT. To achieve good performance, it utilizes several advanced training schemes for convolutional networks. ConvNeXt-Tiny refers to the smallest model size in the ConvNeXt family.

## E HYPERPARAMETERS

In Tables 9 and 10, we list all the key hyperparameters in our experiments for each dataset. Our Appendix accompanies some trained checkpoints and one can easily reproduce.

Table 9: The best hyperparameter of our main experiments (ImageNet-1k).

| ImageNet-1k | ResNet-18 | Inception-V3 | Vision Transformer | ConvNeXt-Tiny |
|---|---|---|---|---|
| # of parallel AC-FNO blocks ($m$) | 2 | 2 | 2 | 4 |
| # of stages ($n$) | 1 | 1 | 1 | 2 |
| First phase training lr. | 1e-3 | 1e-3 | 1e-3 | 2e-4 |
| Second phase training lr. | 1e-6 | 1e-6 | 1e-6 | 2e-6 |

Table 10: The best hyperparameter of our experiments on the fine-grained datasets.

| ConvNeXt-Tiny | StanfordCars | Oxford-IIIT Pets | Flowers | FGVC Aircraft | Food-101 |
|---|---|---|---|---|---|
| # of parallel AC-FNO blocks ($m$) | 4 | 2 | 2 | 2 | 2 |
| # of stages ($n$) | 2 | 2 | 1 | 1 | 1 |
| First phase training lr. | 1e-3 | 1e-3 | 1e-3 | 1e-3 | 1e-3 |
| Second phase training lr. | 1e-6 | 1e-5 | 1e-6 | 1e-6 | 1e-6 |

# F EVALUATION

## F.1 EXPERIMENTAL SETUP

We run our experiments on a machine equipped with Intel i9 CPUs and Nvidia RTX A5000/A6000 GPUs. We implement PAC-FNO using Python v3.8 and PyTorch v1.12.

## F.2 NUMBER OF PARAMETERS OF PAC-FNO

We report the number of parameters of PAC-FNO according to backbone models and datasets.

Table 11: The number of parameters according to backbone models.

| ImageNet-1k | ResNet-18 | Inception-V3 | Vision Transformer | ConvNeXt-Tiny |
|---|---|---|---|---|
| # of PAC-FNO parameters | +0.91M | +1.61M | +0.91M | +3.65M |

Table 12: The number of parameters according to datasets

| ConvNeXt-Tiny | StanfordCars | Oxford-IIIT Pets | Flowers | FGVC Aircraft | Food-101 |
|---|---|---|---|---|---|
| # of PAC-FNO parameters | +3.65M | +3.65M | +3.65M | +3.65M | +3.65M |

## F.3 ADDITIONAL SUPER-RESOLUTION METHODS

In this section, we report experimental results with the additional latest super-resolution baseline. We fine-tune a pre-trained classification model with low-resolution images that are upscaled by the super-resolution model. We note that we used the latest super-resolution model.

In Table 13, combining the super-resolution and fine-tune methods shows better performance than simply upscaling low-resolution images to the target resolution using the super-resolution model. Even at 32 resolution, it shows slightly better performance than PAC-FNO. However, this super-resolution method has two major drawbacks. The first drawback is the model size. DRPN's ×8 upscaling model has a similar number of parameters the the pre-trained model size, i.e., 23.21M vs.28.59M. Second, a model is needed for each resolution. In other words, upscaling models for ×8, ×4, and ×2 are needed to handle 28, 56, and 112 resolutions, respectively. In contrast, our proposed PAC-FNO can handle images of all resolutions with an additional 3.65M network and shows good performance.

Additionally, we verify the superiority of PAC-FNO by providing a comparison with a method equipped with the latest super-resolution model. OSRT (Yu et al., 2023) is a state-of-the-art model in the super-resolution domain but it does not support ×8 upscale. Therefore, we only use the ×2 and ×4 upscale models of OSRT. As a result, PAC-FNO shows better performance than OSRT and OSRT (fine-tune) methods.

Table 13: **Results with the additional latest super-resolution method.** For the experiment, we used ConvNeXt-Tiny as a pre-trained model. (Left: ImageNet-1k, Right: ImageNet-C/P Fog)

| Model | Method | Metric | Resolution 28 | 32 | 56 | 64 | 112 | 128 | 224 |
|---|---|---|---|---|---|---|---|---|---|
| ConvNeXt-Tiny | Resize | Top1-Acc (%) | 27.5 | 34.5 | 60.7 | 64.9 | 75.0 | 77.3 | 82.5 |
| | Fine-tune | Top1-Acc (%) | 40.2 | 62.3 | 65.8 | 76.0 | 76.4 | 80.7 | 81.8 |
| | DRPN | Top1-Acc (%) | 40.7 | - | 68.2 | - | 79.4 | - | 82.5 |
| | | # of Parameters (M) | 23.21 | - | 10.43 | - | 5.95 | - | - |
| | DRPN (Fine-tune) | Top1-Acc (%) | 60.8 | - | 72.5 | - | 76.7 | - | 82.5 |
| | | # of Parameters (M) | 23.21 | - | 10.43 | - | 5.95 | - | - |
| | OSRT | Top1-Acc (%) | - | - | 61.4 | - | 75.4 | - | 82.5 |
| | | # of Parameters (M) | - | - | 11.93 | - | 11.79 | - | - |
| | OSRT (Fine-tune) | Top1-Acc (%) | - | - | 71.2 | - | 78.4 | - | 81.2 |
| | | # of Parameters (M) | - | - | 11.93 | - | 11.79 | - | - |
| | PAC-FNO | Top1-Acc (%) | 58.9 | 63.2 | 77.6 | 76.2 | 80.2 | 80.7 | 81.8 |
| | | # of Parameters (M) | 3.65 | | | | | | |

| Model | Method | Metric | Resolution 28 | 32 | 56 | 64 | 112 | 128 | 224 |
|---|---|---|---|---|---|---|---|---|---|
| ConvNeXt-Tiny | Resize | Top1-Acc (%) | 8.12 | 10.8 | 27.2 | 31.9 | 48.5 | 52.3 | 58.4 |
| | Fine-tune | Top1-Acc (%) | 23.2 | 28.2 | 47.5 | 51.4 | 61.0 | 62.2 | 63.0 |
| | DRPN | Top1-Acc (%) | 0.67 | - | 0.99 | - | 1.32 | - | 58.4 |
| | | # of Parameters (M) | 23.21 | - | 10.43 | - | 5.95 | - | - |
| | DRPN (Fine-tune) | Top1-Acc (%) | 21.8 | - | 42.3 | - | 56.8 | - | 61.0 |
| | | # of Parameters (M) | 23.21 | - | 10.43 | - | 5.95 | - | - |
| | OSRT | Top1-Acc (%) | - | - | 19.4 | - | 37.9 | - | 58.4 |
| | | # of Parameters (M) | - | - | 11.93 | - | 11.79 | - | - |
| | OSRT (Fine-tune) | Top1-Acc (%) | - | - | 42.3 | - | 56.4 | - | 59.4 |
| | | # of Parameters (M) | - | - | 11.93 | - | 11.79 | - | - |
| | PAC-FNO | Top1-Acc (%) | 25.4 | 30.4 | 48.2 | 51.7 | 60.1 | 61.4 | 62.8 |
| | | # of Parameters (M) | 3.65 | | | | | | |

## F.4 Additional fine-grained datasets

Tables 14 and 15 show the performance of the remaining fine-grained dataset, Stanford Cars. PAC-FNO shows good performance in most cases, especially at low-resolution compared to other models. Tables 16 and 17 show the performance of the Oxford-IIIT Pets with the ViT model. PAC-FNO shows good performance than FNO at all resolutions.

Table 14: Top-1 accuracy on low-resolution.

| Dataset | Method | Resolution | | | | | | |
|---|---|---|---|---|---|---|---|---|
| | | 28 | 32 | 56 | 64 | 112 | 128 | **224** |
| Standford Cars | Resize | 14.7 | 22.5 | 66.3 | 73.6 | 88.2 | 89.6 | 91.5 |
| | Fine-tune | 12.3 | 66.5 | 70.5 | 88.5 | 91.2 | 92.4 | 92.6 |
| | DRLN | 1.98 | - | 36.1 | - | 87.0 | - | 91.5 |
| | DRPN | 41.5 | - | 84.2 | - | 90.9 | - | 91.5 |
| | FNO | 11.9 | 19.2 | 67.8 | 75.3 | 91.0 | 92.6 | **93.9** |
| | UNO | 57.4 | 66.2 | 85.9 | 87.8 | 91.6 | 92.1 | 92.3 |
| | A-FNO | 67.6 | 74.4 | 88.0 | 89.7 | 92.1 | 92.3 | 92.7 |
| | PAC-FNO | **70.4** | **76.6** | **89.0** | **90.0** | **92.6** | **92.8** | 93.5 |

Table 15: Relative accuracy on low-resolution.

| Dataset | Method | Resolution | | | | | | |
|---|---|---|---|---|---|---|---|---|
| | | 28 | 32 | 56 | 64 | 112 | 128 | **224** |
| Standford Cars | Resize | 16.1 | 24.5 | 72.4 | 80.4 | 96.4 | 97.9 | 100 |
| | Fine-tune | 13.6 | 71.9 | 76.1 | 95.6 | 98.5 | **99.8** | 100 |
| | DRLN | 2.16 | - | 39.4 | - | 95.1 | - | 100 |
| | DRPN | 51.9 | - | 92.1 | - | 99.3 | - | 100 |
| | FNO | 12.7 | 20.5 | 72.2 | 80.2 | 96.9 | 98.6 | 100 |
| | UNO | 62.2 | 71.7 | 93.1 | 95.1 | 99.2 | **99.8** | 100 |
| | A-FNO | 72.9 | 80.2 | 94.9 | **96.7** | **99.4** | 99.5 | 100 |
| | PAC-FNO | **75.3** | **81.9** | **95.2** | 96.3 | 99.0 | 99.3 | 100 |

Table 16: Top-1 accuracy on low-resolution.

| Dataset | Method | Resolution | | | | | | |
|---|---|---|---|---|---|---|---|---|
| | | 28 | 32 | 56 | 64 | 112 | 128 | **224** |
| Oxford-IIIT Pets | FNO | 26.3 | 33.0 | 58.1 | 64.8 | 82.9 | 85.8 | 91.3 |
| | PAC-FNO | **40.3** | **46.2** | **69.0** | **72.5** | **86.8** | **89.2** | **92.2** |

Table 17: Relative accuracy on low-resolution.

| Model | Method | Resolution | | | | | | |
|---|---|---|---|---|---|---|---|---|
| | | 28 | 32 | 56 | 64 | 112 | 128 | **224** |
| Oxford-IIIT Pets | FNO | 28.8 | 36.1 | 63.6 | 71.0 | 90.8 | 94.0 | 100 |
| | PAC-FNO | **43.7** | **50.1** | **74.8** | **78.6** | **94.1** | **96.7** | **100** |

## F.5 Additional metric for ImageNet-1k and fine-grained datasets

Table 18 and 19 show the relative accuracy of low-resolution images generated by ImageNet-1k and fine-grained datasets. In ImageNet-1k, the performance of PAC-FNO is the best in most of the datasets, and especially in lower resolution. In fine-grained datastes, PAC-FNO shows good performance in all cases. In other words, PAC-FNO works very well for low-resolution image classification.

Table 18: Relative accuracy on low-resolution images generated from ImageNet-1k

| Model | Method | Resolution | | | | | | | |
|---|---|---|---|---|---|---|---|---|---|
| | | 28 | 32 | 56 | 64 | 112 | 128 | **224** | 299 |
| ResNet-18 | Resize | 23.9 | 31.7 | 65.5 | 72.3 | 91.3 | 93.8 | 100 | - |
| | Fine-tune | 1.6 | 3.3 | 15.7 | 25.1 | 53.1 | 77.8 | 100 | - |
| | DRLN | 0.3 | - | 24.5 | - | 90.0 | - | 100 | - |
| | DRPN | 45.3 | - | 79.7 | - | 96.7 | - | 100 | - |
| | FNO | 57.3 | 64.5 | 84.3 | 87.7 | 96.4 | 97.7 | 100 | - |
| | UNO | 57.5 | 65.3 | 84.9 | 88.3 | 96.7 | 98.1 | 100 | - |
| | A-FNO | **63.2** | **72.3** | **87.9** | **91.2** | 97.7 | **98.5** | 100 | - |
| | PAC-FNO | 60.8 | 67.9 | 86.2 | 89.5 | 97.3 | 98.4 | 100 | - |
| Inception-V3 | Resize | 21.6 | 28.5 | 62.9 | 69.7 | 89.9 | 92.9 | - | 100 |
| | Fine-tune | 51.1 | 60.9 | 82.2 | 90.1 | 94.1 | 94.6 | - | 100 |
| | FNO | 62.4 | 68.9 | 87.4 | 89.5 | 95.5 | 97.6 | - | 100 |
| | UNO | 54.6 | 62.2 | 84.6 | 88.0 | 96.6 | 97.7 | - | 100 |
| | A-FNO | 44.6 | 52.9 | 79.6 | 83.7 | 94.8 | 96.3 | - | 100 |
| | PAC-FNO | **62.9** | **70.0** | **87.8** | **90.2** | **97.1** | **98.1** | - | 100 |
| ViT-B16 | Resize | 51.3 | 58.4 | 81.4 | 85.0 | 94.7 | 96.3 | 100 | - |
| | Fine-tune | 49.1 | 59.8 | 82.1 | 85.8 | 95.3 | 96.7 | 100 | - |
| | DRLN | 4.6 | - | 51.8 | - | 94.9 | - | 100 | - |
| | DRPN | 64.4 | - | 90.0 | - | **98.6** | - | 100 | - |
| | FNO | 49.1 | 58.5 | 83.1 | 86.8 | 96.7 | 97.9 | 100 | - |
| | UNO | 54.5 | 63.1 | 84.7 | 87.7 | 97.1 | 98.3 | 100 | - |
| | A-FNO | **69.1** | **74.8** | **90.9** | **93.2** | 98.5 | **99.2** | 100 | - |
| | PAC-FNO | 58.5 | 65.5 | 86.7 | 89.9 | 97.6 | 98.1 | 100 | - |
| ConvNeXt-Tiny | Resize | 33.3 | 41.8 | 73.6 | 78.7 | 90.9 | 93.7 | 100 | - |
| | Fine-tune | 49.1 | 76.2 | 80.4 | 92.9 | 93.4 | 98.7 | 100 | - |
| | DRLN | 0.4 | - | 30.4 | - | 87.0 | - | 100 | - |
| | DRPN | 49.3 | - | 82.7 | - | 96.2 | - | 100 | - |
| | FNO | 54.0 | 61.7 | 85.1 | 88.1 | 96.6 | 97.7 | 100 | - |
| | UNO | 72.7 | **78.0** | **91.4** | 93.7 | 98.1 | 98.9 | 100 | - |
| | A-FNO | 61.8 | 68.5 | 87.2 | 90.3 | 97.2 | 98.1 | 100 | - |
| | PAC-FNO | **72.3** | 77.5 | **91.4** | **93.5** | **98.4** | 99.0 | 100 | - |

Table 19: Relative accuracy on low-resolution images generated from Fine-grained datasets

| Dataset | Method | Resolution | | | | | | |
|---|---|---|---|---|---|---|---|---|
| | | 28 | 32 | 56 | 64 | 112 | 128 | **224** |
| Oxford-IIIT Pets | Resize | 31.4 | 38.5 | 74.8 | 82.2 | 95.9 | 97.3 | 100 |
| | Fine-tune | 34.8 | 43.8 | 78.0 | 84.5 | 97.0 | 97.5 | 100 |
| | DRLN | 3.6 | - | 39.3 | - | 94.0 | - | 100 |
| | DRPN | 44.3 | - | 89.9 | - | 98.8 | - | 100 |
| | FNO | 20.9 | 59.1 | 66.3 | 76.8 | 94.9 | 98.9 | 100 |
| | UNO | 12.3 | 17.4 | 47.8 | 55.9 | 88.7 | 92.8 | 100 |
| | A-FNO | 30.3 | 37.4 | 71.3 | 78.8 | 95.6 | 97.5 | 100 |
| | PAC-FNO | **80.0** | **84.3** | **93.8** | **95.5** | **98.7** | **99.3** | 100 |
| Flowers | Resize | 41.2 | 49.6 | 78.5 | 83.7 | 96.0 | 97.7 | 100 |
| | Fine-tune | 46.2 | 53.3 | 82.2 | 85.4 | 96.5 | 98.0 | 100 |
| | DRLN | 10.3 | - | 55.7 | - | 95.2 | - | 100 |
| | DRPN | 67.3 | - | **92.8** | - | 99.1 | - | 100 |
| | FNO | 26.2 | 34.0 | 68.2 | 75.6 | 94.9 | 97.0 | 100 |
| | UNO | 24.1 | 31.4 | 67.1 | 74.6 | 94.5 | 96.4 | 100 |
| | A-FNO | 29.2 | 36.6 | 69.3 | 76.2 | 93.8 | 96.6 | 100 |
| | PAC-FNO | **78.5** | **82.6** | **92.8** | **94.6** | **99.2** | **99.8** | 100 |
| FGVC Aircraft | Resize | 3.1 | 3.5 | 34.3 | 52.4 | 89.1 | 93.2 | 100 |
| | Fine-tune | 10.7 | 20.3 | 50.4 | 73.1 | **96.7** | 95.2 | 100 |
| | DRLN | 1.4 | - | 16.4 | - | 86.4 | - | 100 |
| | DRPN | 23.1 | - | 74.8 | - | 96.6 | - | 100 |
| | FNO | 32.6 | 41.1 | 73.6 | 79.3 | 92.4 | 93.8 | 100 |
| | UNO | 2.0 | 2.1 | 11.2 | 27.8 | 89.5 | 93.6 | 100 |
| | A-FNO | 8.4 | 14.2 | 62.9 | 73.3 | 92.9 | 95.3 | 100 |
| | PAC-FNO | **46.5** | **55.6** | **82.6** | **85.6** | 94.9 | **96.9** | 100 |
| Food-101 | Resize | 44.0 | 52.8 | 83.6 | 88.0 | 96.7 | 97.9 | 100 |
| | Fine-tune | 55.0 | 59.3 | 88.0 | 89.9 | 96.8 | 98.2 | 100 |
| | DRLN | 7.7 | - | 42.9 | - | 95.3 | - | 100 |
| | DRPN | 60.2 | - | 89.6 | - | 65.3 | - | 100 |
| | FNO | 48.0 | 56.5 | 85.8 | 89.2 | 97.1 | 98.0 | 100 |
| | UNO | 57.5 | 64.5 | 85.9 | 89.1 | 96.9 | 97.8 | 100 |
| | A-FNO | 52.2 | 58.6 | 81.9 | 85.9 | 95.0 | 96.1 | 100 |
| | PAC-FNO | **82.9** | **86.1** | **94.6** | **96.1** | **98.7** | **99.2** | 100 |

## F.6 ADDITIONAL NATURAL INPUT VARIATIONS

Table 20 shows the remaining input variation results of ImageNet-C/P Hendrycks & Dietterich (2019). For the remaining input variations, we report results at 32, 64, 128, and 224 resolution, excluding SR models whose performance appears to be meaningless. In most cases, PAC-FNO shows good performance at 32 and 64 resolution, and Fine-tune shows good performance at 128×128 and 224×224 resolution. However, at 128 and 224, there is a performance difference of up to 5% (Glass noise 44.2% vs. 39.5% at 128 × 128 resolution), but at 32 and 63, there is a performance difference of up to 47% (Pixelate 44.2% vs. 39.5% at 32 × 32 resolution). In other words, PAC-FNO shows good performance by a large margin at low-resolution.

Table 20: **Performance of PAC-FNO on the input variation tasks.** We show the top-1 accuracy of the ConvNext-Tiny model on remaining input variations, chosen from ImageNet-C/P (Hendrycks & Dietterich, 2019).

| Variation | Model | Resolution | | | |
|---|---|---|---|---|---|
| | | 32 | 64 | 128 | 224 |
| | Resize | 11.3 | 37.9 | 55.3 | 47.9 |
| | Fine-tune | 11.6 | 39.5 | **57.4** | 51.5 |
| Gaussian | FNO | 38.4 | 47.1 | 46.5 | 43.4 |
| Noise | UNO | 42.8 | 54.6 | 56.5 | 52.1 |
| | A-FNO | 37.8 | 43.1 | 44.0 | 42.8 |
| | PAC-FNO | **48.0** | **54.8** | 56.5 | **52.3** |
| | Resize | 11.3 | 37.3 | 53.8 | 45.4 |
| | Fine-tune | 11.6 | 39.0 | **56.2** | 49.5 |
| Shot | FNO | 38.2 | 45.9 | 43.9 | 41.1 |
| Noise | UNO | 41.2 | 53.6 | 55.2 | **50.5** |
| | A-FNO | 38.1 | 42.9 | 42.8 | 41.2 |
| | PAC-FNO | **47.7** | **54.0** | 55.0 | 50.0 |
| | Resize | 10.8 | 36.9 | 53.2 | 44.6 |
| | Fine-tune | 11.3 | 38.5 | **55.5** | 48.5 |
| Impulse | FNO | 37.2 | 45.2 | 41.8 | 38.5 |
| Noise | UNO | 37.2 | 53.5 | 54.3 | 48.9 |
| | A-FNO | 36.5 | 41.3 | 39.3 | 37.2 |
| | PAC-FNO | **46.7** | **53.5** | 54.1 | **50.2** |
| | Resize | 10.0 | 32.2 | 48.3 | 43.2 |
| | Fine-tune | 11.1 | 36.9 | 56.9 | 55.9 |
| Defocus | FNO | 35.8 | 44.9 | 47.4 | 47.6 |
| Noise | UNO | 47.2 | 51.5 | 50.5 | 47.3 |
| | A-FNO | 38.6 | 47.2 | 47.4 | 47.2 |
| | PAC-FNO | **51.3** | **57.7** | **57.5** | **56.2** |
| | Resize | 11.7 | 33.5 | 38.7 | 31.5 |
| | Fine-tune | 12.7 | 36.9 | **44.2** | **38.8** |
| Glass | FNO | 39.8 | 43.3 | 34.9 | 32.4 |
| Noise | UNO | 26.1 | 43.4 | 33.4 | 30.7 |
| | A-FNO | 42.8 | 39.6 | 28.7 | 26.1 |
| | PAC-FNO | **54.4** | **50.9** | 39.5 | 35.1 |

| Variation | Model | Resolution | | | |
|---|---|---|---|---|---|
| | | 32 | 64 | 128 | 224 |
| | Resize | 10.6 | 36.5 | 54.0 | 48.1 |
| | Fine-tune | 11.5 | 39.6 | **58.8** | **55.9** |
| Motion | FNO | 36.3 | 46.9 | 48.7 | 47.4 |
| Noise | UNO | 41.3 | 50.6 | 49.3 | 47.9 |
| | A-FNO | 37.6 | 44.9 | 42.6 | 41.3 |
| | PAC-FNO | **50.4** | **56.5** | 55.3 | 54.1 |
| | Resize | 10.3 | 25.7 | 43.0 | 43.6 |
| | Fine-tune | 10.9 | 26.8 | 46.4 | **48.9** |
| Zoom | FNO | 35.2 | 38.5 | 37.3 | 37.0 |
| Noise | UNO | 33.5 | 40.2 | 39.4 | 39.5 |
| | A-FNO | 36.0 | 35.9 | 33.8 | 33.5 |
| | PAC-FNO | **49.6** | **48.8** | **46.5** | 44.6 |
| | Resize | 4.9 | 20.2 | 46.6 | 49.6 |
| | Fine-tune | 5.2 | 21.8 | **49.0** | **52.5** |
| Snow | FNO | 20.3 | 31.1 | 41.6 | 45.9 |
| | UNO | 40.1 | 36.5 | 44.0 | 43.6 |
| | A-FNO | 18.1 | 27.4 | 37.5 | 40.4 |
| | PAC-FNO | 31.9 | **41.1** | 48.3 | 49.4 |
| | Resize | 6.5 | 30.8 | 54.9 | 54.1 |
| | Fine-tune | 6.9 | 32.3 | 56.7 | **57.0** |
| Frost | FNO | 27.8 | 42.3 | 49.0 | 50.9 |
| | UNO | 46.0 | 48.8 | 54.0 | 53.6 |
| | A-FNO | 22.6 | 37.7 | 45.3 | 46.0 |
| | PAC-FNO | **40.3** | **52.4** | **56.9** | 56.9 |
| | Resize | 8.1 | 40.4 | 62.9 | 59.5 |
| | Fine-tune | 8.5 | 41.7 | **65.8** | 64.7 |
| Contrast | FNO | 31.8 | 53.5 | 63.3 | 63.5 |
| | UNO | 59.2 | 59.6 | 65.2 | 62.8 |
| | A-FNO | 34.8 | 55.6 | 62.1 | 59.2 |
| | PAC-FNO | 44.9 | **59.9** | 65.1 | 62.3 |

| Variation | Model | Resolution | | | |
|---|---|---|---|---|---|
| | | 32 | 64 | 128 | 224 |
| | Resize | 12.2 | 40.0 | 56.8 | 53.2 |
| | Fine-tune | 12.6 | 41.5 | **59.1** | **56.6** |
| Elastic | FNO | 38.3 | 49.7 | 52.1 | 51.5 |
| Transform | UNO | 45.4 | 52.8 | 50.5 | 48.4 |
| | A-FNO | 41.3 | 48.2 | 47.8 | 45.4 |
| | PAC-FNO | **51.8** | **57.3** | 55.8 | 53.5 |
| | Resize | 14.5 | 81.0 | 63.8 | 53.2 |
| | Fine-tune | 15.1 | 52.9 | 66.8 | **56.6** |
| Pixelate | FNO | 48.6 | 66.7 | 62.5 | 42.9 |
| | UNO | 40.8 | 70.9 | 64.6 | 52.2 |
| | A-FNO | 53.2 | 67.8 | 58.9 | 40.8 |
| | PAC-FNO | **62.6** | **73.2** | **67.9** | 54.4 |
| | Resize | 12.9 | 44.4 | 64.3 | 62.4 |
| | Fine-tune | 13.3 | 45.9 | 66.3 | 65.5 |
| Jpeg | FNO | 45.2 | 62.5 | **65.2** | **63.7** |
| Compression | UNO | **61.1** | 61.2 | 64.4 | 62.2 |
| | A-FNO | 50.9 | **64.8** | 63.9 | 61.1 |
| | PAC-FNO | 52.3 | 63.2 | 64.8 | 63.0 |
| | Resize | 12.1 | 41.2 | 59.2 | 53.5 |
| | Fine-tune | 12.5 | 42.6 | **66.2** | 57.1 |
| Speckle | FNO | 41.6 | 52.7 | 51.3 | 48.8 |
| Noise | UNO | 48.3 | 58.6 | 59.9 | 56.2 |
| | A-FNO | 42.7 | 49.6 | 49.8 | 48.3 |
| | PAC-FNO | **52.1** | **60.3** | 61.4 | **57.8** |
| | Resize | 10.4 | 34.8 | 51.1 | 46.9 |
| | Fine-tune | 11.5 | 39.1 | 59.0 | 58.4 |
| Gaussian | FNO | 37.3 | 47.8 | 50.6 | 50.8 |
| Blur | UNO | 51.2 | 54.0 | 53.3 | 50.2 |
| | A-FNO | 40.1 | 50.2 | 51.3 | 51.2 |
| | PAC-FNO | **52.5** | **60.0** | **60.7** | **59.4** |

Table 21: **Performance of PAC-FNO according to low and high-frequency filter.** We report top-1 accuracy on low-resolution images generated from ImageNet-1k in ConvNeXt-Tiny.

| ImageNet-1k | 32 | 64 | 128 | 224 |
|---|---|---|---|---|
| PAC-FNO (low pass filter) | 53.5 | 71.4 | 78.7 | 79.0 |
| PAC-FNO (high pass filter) | 21.6 | 49.4 | 68.2 | 74.8 |
| PAC-FNO | **58.9** | **74.5** | **80.2** | **81.5** |

| ImageNet-C/P Fog | 32 | 64 | 128 | 224 |
|---|---|---|---|---|
| PAC-FNO (low pass filter) | 18.0 | 41.7 | 52.4 | 54.4 |
| PAC-FNO (high pass filter) | 5.92 | 23.0 | 43.2 | 50.2 |
| PAC-FNO | **25.4** | **48.2** | **60.1** | **62.8** |

## F.7 ADDITIONAL ABLATION STUDIES

We provide an analysis of the impact of low and high-frequency information on accuracy/generalization through ablation experiments. Table 21 shows thah compared to PAC-FNO, using low-pass and high-pass filters results in lower accuracy and generalization. When using a high-pass filter, it is expected to show good performance in ImageNet-C/P Fog, but since the perfor-

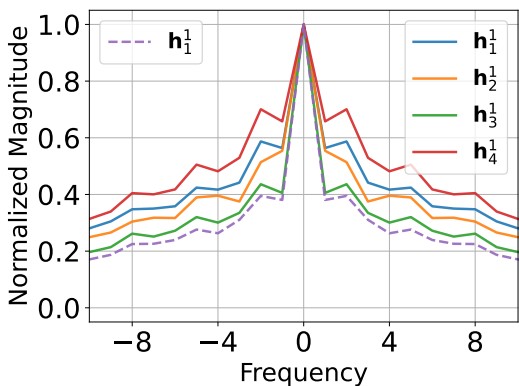

Figure 7: **Comparison of spectral responses according to the configuration of the AC-FNO block.** We test the ConvNeXT-Tiny backbone model on ImageNet-1k and visualize only the hidden vector of the first layer ($m = 1$) for the hidden vectors $\mathbf{h}_n^m$. In the case of parallel (solid line), there are four hidden vectors ($\mathbf{h}_n^1, n \in \{1, 2, 3, 4\}$), and in the case of serial (dashed line), there is one hidden vector ($\mathbf{h}_n^1, n \in \{1\}$).

mance even on clean images is not good, it does not show good performance in terms of generalization. Therefore, only PAC-FNO, which uses both low and high-frequency components, shows good performance in terms of accuracy/generalization.

### F.8    EFFECTIVENESS OF PARALLEL ARCHITECTURE

In this section, we show the efficacy of the parallel configuration of the AC-FNO block by visualizing which frequencies are captured. For fair comparison, we visualize the first layer output, which contains the most information of an original input sample, for the following two settings: AC-FNO in our proposed parallel configuration and AC-FNO in a serial configuration. Figure 7 shows spectral responses according to the configuration of the AC-FNO block. The farther it is from the center, the higher its frequency is.

In Figure 7, We show that in the parallel configuration, each hidden vector not only captures the low-frequency components but also captures the high-frequency components. Moreover, their frequencies are sometimes complementary to each other. In particular, $\mathbf{h}_4^1$ has large normalized magnitudes at high frequency ranges, which means that $\mathbf{h}_3^1$ captures high-frequency components well. On the other hand, the hidden vector of the serial configuration (dashed line) is concentrated at low-frequency (center).

Additionally, since the parallel configuration of the AC-FNO block captures both high and low-frequency components, it also shows good performance for image degradations as shown in Figure 8. Figures 8d and 8e are visualizations in the frequency domain. In other words, PAC-FNO consider high-frequency information well in those cases.

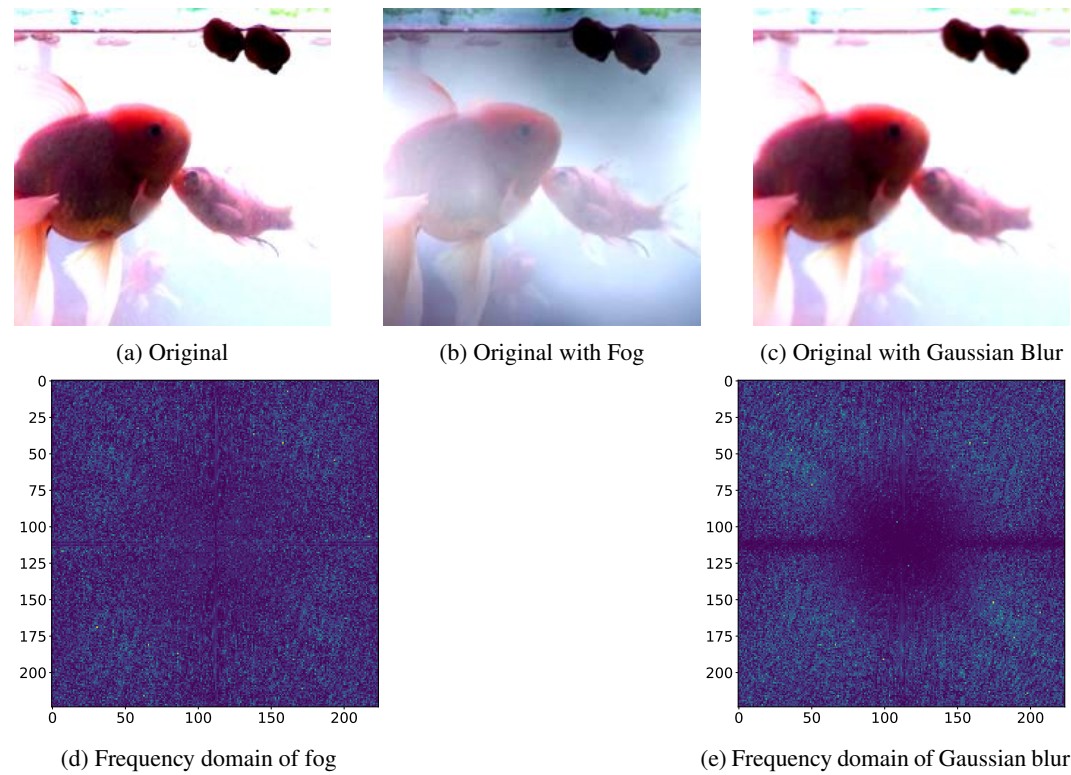

(a) Original      (b) Original with Fog      (c) Original with Gaussian Blur

(d) Frequency domain of fog      (e) Frequency domain of Gaussian blur

Figure 8: **Visualization of image degradation.** (d) and (e) are visualizations of degradation without clean images e.g., (b)-(a) and (c)-(a) in the frequency domain.

## F.9 FLOPs AND RUNTIME

We report the FLOPs and runtime on data at different resolutions

Table 22: Results of FLOPs and runtime on ImageNet-1k in ConvNeXt-Tine.

| ImageNet-1k | Method | Metrics | Resolution | | | |
| --- | --- | --- | --- | --- | --- | --- |
| | | | 28 | 56 | 112 | 224 |
| ConvNeXt-Tiny | Resize | GFLOPs | 8.96 | 8.96 | 8.96 | 8.96 |
| | | Runtimes (s) | 0.006 | 0.006 | 0.006 | 0.006 |
| | Fine-tune | GFLOPs | 8.96 | 8.96 | 8.96 | 8.96 |
| | | Runtimes (s) | 0.006 | 0.006 | 0.006 | 0.006 |
| | DRLN | GFLOPs | 180.66 | 412.05 | 1200.50 | 8.96 |
| | | Runtimes (s) | 0.378 | 0.498 | 0.913 | 0.007 |
| | DRPN | GFLOPs | 1220.42 | 576.94 | 387.88 | 8.96 |
| | | Runtimes (s) | 0.1532 | 0.164 | 0.171 | 0.007 |
| | FNO | GFLOPs | 9.78 | 9.78 | 9.78 | 9.78 |
| | | Runtimes (s) | 0.016 | 0.016 | 0.016 | 0.016 |
| | UNO | GFLOPs | 9.10 | 9.10 | 9.10 | 9.10 |
| | | Runtimes (s) | 0.018 | 0.018 | 0.018 | 0.018 |
| | AFNO | GFLOPs | 8.96 | 8.96 | 8.96 | 8.96 |
| | | Runtimes (s) | 0.010 | 0.010 | 0.010 | 0.010 |
| | PAC-FNO | GFLOPs | 8.98 | 8.98 | 8.98 | 8.98 |
| | | Runtimes (s) | 0.013 | 0.013 | 0.013 | 0.013 |

