# OpenReview forum: "PAC-FNO: Parallel-Structured All-Component Fourier Neural Operators for Recognizing Low-Quality Images"
_ICLR.cc/2024/Conference — ICLR 2024 poster_

### Official Review · Reviewer_pDdt · 2023-10-30

**Soundness:** 3 good
**Presentation:** 2 fair
**Contribution:** 2 fair
**Rating:** 6
**Confidence:** 5

**Summary:**

The paper presents a novel solution named PAC-FNO for image recognition, demonstrating the ability to simultaneously handle images of varying resolutions and resist the impact of various types of input-induced natural variations within a singular model in recognition tasks. The proposed parallel-structured and all-component Fourier neural operator (PAC-FNO), building on the resolution invariance of FNOs in the frequency domain, notably eliminates the ideal low-pass filter found in vanilla FNOs. Additionally, it transforms the traditional serial architectures into a parallel structure, thereby considering a broader range of frequency components, retaining high-frequency details, and notably enhancing performance, especially in fine-grained datasets. The proposed approach introduces a two-stage training method that fine-tunes pre-trained image recognition models in conjunction with PAC-FNO, allowing the acquisition of commonalities among various input resolutions with minimal modifications to the backbone classification network. Through conducted experiments, the authors effectively showcase the performance of PAC-FNO, significantly improving accuracy in comparison to existing baseline models. The manuscript is well-written, and the experiments conducted are comprehensive and convincingly articulated.

**Strengths:**

1. The paper exhibits a high level of innovation. Although neural network operators based on the Fourier domain transformation for learning, due to their excellent characteristics in resolution invariance, have been recognized and applied in various areas, especially for enhancing pre-processing operators in variable-resolution input networks. However, the authors, motivated by the rational desire to retain high-frequency image details, proposed for the first time to eliminate the inherent low-pass filters in the model. Additionally, they introduced a popular parallel structure similar to Multi-head Self-Attention, further enhancing the network's performance while expanding the design philosophy of relevant operators.

2. The presentation of this paper is professional and fluent. It has almost no expression errors and clearly elucidates the authors' contributions.

3. The paper conducted extensive and meticulous experiments, utilizing seven image recognition benchmark datasets and applying the operators to four different backbone networks. The authors closely follow the cutting-edge developments in the field, employing more advanced VIT and ConvNeXt for experimentation, which makes the results highly persuasive.

**Weaknesses:**

1. Although the author compared super-resolution (SR) models for variable resolution inputs, the compared SR models are outdated and lack representation across various upscaling factors for super-resolution reconstruction. The field of super-resolution has seen significant advancements recently; thus, it is recommended to select more appropriate comparative algorithms.

2. The primary advantage of Fourier Neural Operators (FNOs) lies in their use of frequency domain processing for resolution invariance. As a learnable enhancement operator, it's expected to exhibit some resilience to input natural variations. However, the author hasn't provided a detailed and explanatory analysis of the mechanisms where the operator shows robustness against natural variations. Moreover, the chosen input variations in the experiments, like fog, brightness, spatter, and saturate, represent basic degradation scenarios that can be addressed without deep learning methods. Therefore, regarding resilience to input natural variations, this might not be sufficiently emphasized as a highlight of the paper. The paper suggests exploring degradation in real-world scenarios in future work, indicating that the authors are aware of the limitations in terms of experimental performance or the algorithm proposed. However, such scenarios represent fundamental problems studied in the field of Image Recognition (IR) and hold significant practical application implications. Actually, certain degradation processes might affect high or low frequency details in the image's frequency domain. For instance, blur involves the loss of high-frequency details, prompting the author to conduct a mechanistic analysis combining frequency domain and degradation processes to enhance this aspect's interpretability.

3. The original intent behind the existence of ideal low-pass filters was to reduce the number of parameters and computational complexity. While the author's innovative design to remove the inherent low-pass filter is intuitively comprehensible, the associated trade-offs are not discussed in the manuscript. It would be beneficial to provide supplementary explanations to demonstrate the worthiness of such a modification.

4. The experiments thoroughly prove the advantages of parallel architectures and claim that this approach encapsulates more frequency components. However, they lack further detailed explanations and justifications.

**Questions:**

See Weaknesses.

---

> ### Author Response · Authors · 2023-11-17
>
> Thank you for the encouraging remarks about our contribution and extensive experimental results and the valuable feedback from the reviewer. We hope that our responses will solve the reviewer’s questions.
>
> > 1. Although the author compared super-resolution (SR) models for variable resolution inputs, the compared SR models are outdated and lack representation across various upscaling factors for super-resolution reconstruction. The field of super-resolution has seen significant advancements recently; thus, it is recommended to select more appropriate comparative algorithms.
>
> $\to$ As suggested, we conduct the experiments of other methods with a state of the art SR model and summarize the results in the revision (will update it by  November 20th (AOE)).
>
> > 2. The primary advantage of Fourier Neural Operators (FNOs) lies in their use of frequency domain processing for resolution invariance. As a learnable enhancement operator, it's expected to exhibit some resilience to input natural variations. However, the author hasn't provided a detailed and explanatory analysis of the mechanisms where the operator shows robustness against natural variations. Moreover, the chosen input variations in the experiments, like fog, brightness, spatter, and saturate, represent basic degradation scenarios that can be addressed without deep learning methods. Therefore, regarding resilience to input natural variations, this might not be sufficiently emphasized as a highlight of the paper. The paper suggests exploring degradation in real-world scenarios in future work, indicating that the authors are aware of the limitations in terms of experimental performance or the algorithm proposed. However, such scenarios represent fundamental problems studied in the field of Image Recognition (IR) and hold significant practical application implications. Actually, certain degradation processes might affect high or low frequency details in the image's frequency domain. For instance, blur involves the loss of high-frequency details, prompting the author to conduct a mechanistic analysis combining frequency domain and degradation processes to enhance this aspect's interpretability.
>
> $\to$ As you mentioned, degradations change the high-frequency and low-frequency information in original images. Figure 8 in the Appendix F.8 shows that there are many changes in high-frequency when degradations are visualized in the frequency domain. Since existing FNO-based models that use a low-pass filter remove high-frequency, the degraded image loses not only the degradation factor but also the information of the original image. However, our proposed PAC-FNO uses high-frequency information, so it shows better performance in degradation.
> Regarding the future work, we would like to address combined degradation types as the natural variations (e.g., mixed with motion blur and fog that may occur in the real-world), not try to address new types of variations.

---

> ### Author Response · Authors · 2023-11-17
>
> > 3. The original intent behind the existence of ideal low-pass filters was to reduce the number of parameters and computational complexity. While the author's innovative design to remove the inherent low-pass filter is intuitively comprehensible, the associated trade-offs are not discussed in the manuscript. It would be beneficial to provide supplementary explanations to demonstrate the worthiness of such a modification.
>
> $\to$ We provide FLOPs and runtime on data at different resolutions. Existing FNO models used low-pass filters due to computational complexity, but the channel size was increased to fully utilize low-frequency components. However, PAC-FNO maintained the 3 channel size of the image to use both high-pass filter and low-pass filter. As a result, PAC-FNO showed similar levels of FLOPs and Runtimes as existing FNO models, but showed better performance because it used additional high-frequency components.
>
> | ImageNet-1k |    Metrics   |    28   |   56   |   112   |  224  |
> |:-----------:|:------------:|:-------:|:------:|:-------:|:-----:|
> |    Resize   |    GFLOPs    |   8.96  |  8.96  |   8.96  |  8.96 |
> |             | Runtimes (s) |  0.006  |  0.006 |  0.006  | 0.006 |
> |  Fine-tune  |    GFLOPs    |   8.96  |  8.96  |   8.96  |  8.96 |
> |             | Runtimes (s) |  0.006  |  0.006 |  0.006  | 0.006 |
> |     DRLN    |    GFLOPs    |  180.66 | 412.05 | 1200.50 |  8.96 |
> |             | Runtimes (s) |  0.378  |  0.498 |  0.913  | 0.007 |
> |     DRPN    |    GFLOPs    | 1220.42 | 576.94 |  387.88 |  8.96 |
> |             | Runtimes (s) |  0.1532 |  0.164 |  0.171  | 0.007 |
> |     FNO     |    GFLOPs    |   9.78  |  9.78  |   9.78  |  9.78 |
> |             | Runtimes (s) |  0.016  |  0.016 |  0.016  | 0.016 |
> |     UNO     |    GFLOPs    |   9.10  |  9.10  |   9.10  |  9.10 |
> |             | Runtimes (s) |  0.018  |  0.018 |  0.018  | 0.018 |
> |     AFNO    |    GFLOPs    |   8.96  |  8.96  |   8.96  |  8.96 |
> |             | Runtimes (s) |  0.010  |  0.010 |  0.010  | 0.010 |
> |   PAC-FNO   |    GFLOPs    |   8.98  |  8.98  |   8.98  |  8.98 |
> |             | Runtimes (s) |  0.013  |  0.013 |  0.013  | 0.013 |
>
> > 4. The experiments thoroughly prove the advantages of parallel architectures and claim that this approach encapsulates more frequency components. However, they lack further detailed explanations and justifications.
>
> $\to$ The parallel configuration of AC-FNO blocks captures more information than a serial structured model in the first layer, which is directly related to the data. We visualize what frequency the parallel configuration and serial configuration capture in Figure 7 in Appendix F.8 of revision. Figure 7 shows that the parallel configuration captures more high frequencies than the serial configuration. In other words, the parallel structured model captured both more low-frequency and high-frequency components than the serial structured model. As a result, parallel configuration of AC-FNO blocks show a better performance than serial configuration in Figure 4 in Section 4.3. We will add this explanation and justification in Appendix F.8 of the revision. Thank you!

---

> ### Author Response · Authors · 2023-11-20
>
> We provide an additional comparison to the method equipped with the latest super-resolution model. The results are in the following table. OSRT [1] is a state-of-the-art model in the super-resolution domain but it does not support $\times$8 upscale. Therefore, we only use the $\times$2 and $\times$4 upscale models of OSRT. In addition, we report OSRT (fine-tune) that combines super-resolution and fine-tuning methods. That is, we fine-tune the pre-trained classification model with low-resolution images upscaled by a super-resolution model.  As a result, PAC-FNO shows better performance than OSRT and OSRT (fine-tune) methods. We also report these results in Table 13 in Appendix F.3 of the revision.
>
> |  Imagenet-1k  |    Method   |        Metric       |    28    |    32    |    56    |    64    |    112   |    128   |  224 |
> |:-------------:|:-----------:|:-------------------:|:--------:|:--------:|:--------:|:--------:|:--------:|:--------:|:----:|
> |              |     OSRT    |    Top-1 Acc (%)    |     -    |     -    |   61.4   |     -    |   75.4   |     -    | 82.5 |
> |               |             | # of Parameters (M) |     -    |     -    |   11.93  |     -    |   11.93  |     -    |   -  |
> |  ConvNeXt-Tiny  |     OSRT    |    Top-1 Acc (%)    |     -    |     -    |   71.2   |     -    |   78.4   |     -    | 82.1 |
> |               | (Fine-tune) | # of Parameters (M) |     -    |     -    |   11.93  |     -    |   11.79  |     -    |   -  |
> |               |   PAC-FNO   |    Top-1 Acc (%)    |  **58.9**   | **63.2** | **77.6** | **76.2** | **80.2** | **80.7** | 81.8 |
> |              |             | # of Parameters (M) |          |          |          |   3.65   |          |          |      |
>
> | Imagenet-C/P Fog |    Method   |        Metric       |    28    |    32    |    56    |    64    |    112   |    128   |    224   |
> |:----------------:|:-----------:|:-------------------:|:--------:|:--------:|:--------:|:--------:|:--------:|:--------:|:--------:|
> |                 |     OSRT    |    Top-1 Acc (%)    |     -    |     -    |   19.4   |     -    |   37.9   |     -    |   58.4   |
> |                  |             | # of Parameters (M) |     -    |     -    |   11.93  |     -    |   11.93  |     -    |     -    |
> | ConvNeXt-Tiny  |     OSRT    |    Top-1 Acc (%)    |     -    |     -    |   42.3   |     -    |   56.4   |     -    |   59.4   |
> |                  | (Fine-tune) | # of Parameters (M) |     -    |     -    |   11.93  |     -    |   11.79  |     -    |     -    |
> |                  |   PAC-FNO   |    Top-1 Acc (%)    | **25.4** | **30.4** | **48.2** | **51.7** | **60.1** | **61.4** | **62.8** |
> |         `        |             | # of Parameters (M) |          |          |          |   3.65   |          |          |          |
>
> [1] Yu, Fanghua, et al. "OSRT: Omnidirectional image super-resolution with distortion-aware transformer." Proceedings of the IEEE/CVF Conference on Computer Vision and Pattern Recognition. 2023.

---

> ### Author Response · Authors · 2023-11-22
>
> Dear Reviewer pDdt,
>
> We appreciate the reviewer’s time and effort in reviewing our manuscript and insightful comments.
>
> As the closure of the discussion period is approaching, we would like to bring the review’s attention and check if the reviewer could let us know whether the concerns or the misunderstanding have been addressed.
>
> If this is the case, we would appreciate if you could adjust your rating accordingly.
>
> Best regards,
>
> Authors

---

> > ### Comment · Reviewer_pDdt · 2023-11-22
> > **Response**
> >
> > Thanks for your time! My major concerns have already been addressed. However, I have several problems.
> > - Some degradation-resistant algorithms seem to do the same thing as you and I want to see further discussions in your paper to make me better understand your contribution, such as Degradation-Resistant Unfolding Network for Heterogeneous Image Fusion (ICCV 23) and HQG-Net: Unpaired Medical Image Enhancement with High-Quality Guidance (TNNLS).
> >
> > - Although this paper focuses on solving the recognition problem, I'd like to see the author expand it to other extreme tasks, such as camouflaged object segmentation [1, 2] and concealed object segmentation [3]. Perhaps there isn't enough time for the author to continue with the corresponding experiments, but I think it's also more useful to see the author's point of view on a high level for other interested readers to get inspiration from this paper.
> >
> > [1] Camouflaged object detection, CVPR20.
> > [2] Camouflaged Object Detection with Feature Decomposition and Edge Reconstruction, CVPR23.
> > [3] Weakly-Supervised Concealed Object Segmentation with SAM-based Pseudo Labeling and Multi-scale Feature Grouping, NeurIPS23.

---

> ### Author Response · Authors · 2023-11-22
>
> We are happy to hear that your major concern has been resolved. We hope our additional answers will also help you with your problems.
> > 1. Some degradation-resistant algorithms seem to do the same thing as you and I want to see further discussions in your paper to make me better understand your contribution, such as Degradation-Resistant Unfolding Network for Heterogeneous Image Fusion (ICCV 23) and HQG-Net: Unpaired Medical Image Enhancement with High-Quality Guidance (TNNLS).
>
> $\to$ Thank you for pointing out the work. In a nutshell, both papers aim to convert low-quality images into high-quality images.
> [1] generates a high-quality image by fusing low-quality images generated from multiple sensors. [2] generates high-quality medical images using a model trained with unpaired low-quality medical images and high-quality medical images.
>
> However, there is a difference between the two studies and our PAC-FNO. PAC-FNO handles low-quality images including low-resolution and input variations. The low-quality images in these two studies include only input variations such as images with fog or noise. In other words, PAC-FNO handles not only the input variations mentioned in those studies, but also low-resolution images, which occurs more frequently in real world deployment scenarios, *e.g.*, CCTV.
>
> [1] Degradation-Resistant Unfolding Network for Heterogeneous Image Fusion (ICCV 23)
>
> [2] HQG-Net: Unpaired Medical Image Enhancement with High-Quality Guidance (TNNLS)
>
>
> > 2. Although this paper focuses on solving the recognition problem, I'd like to see the author expand it to other extreme tasks, such as camouflaged object segmentation [1, 2] and concealed object segmentation [3]. Perhaps there isn't enough time for the author to continue with the corresponding experiments, but I think it's also more useful to see the author's point of view on a high level for other interested readers to get inspiration from this paper.
>
> $\to$ Very interesting suggestion!  As PAC-FNO addresses the semantic mapping (*i.e.*, classification) problem as it has the most wide applicability, we believe it could be extended to suggested extreme segmentation tasks with a trivial extension. Since the the dense (*i.e.*, pixel-wise) prediction task is essentially the semantic mapping (*i.e.*, classification) problem by taking into account the nearby pixels, given sufficiently large receptive fields tuned by convolution filters for CNNs or token patches for transformers. Empirical validation on this suggestion would be a great future work to widen the applicability of the PAC-FNO. Thank you!

---

> > ### Comment · Reviewer_pDdt · 2023-11-23
> > **Response**
> >
> > The discussions above are excellent, and I wish I could see them in the revised manuscript, which I think would help readers better understand the contributions of this article.
> >
> > Given that my major concerns have already been addressed, I am now happy to update my rating to 6. Best of luck to the authors!

---

### Official Review · Reviewer_WCy8 · 2023-10-31

**Soundness:** 3 good
**Presentation:** 3 good
**Contribution:** 3 good
**Rating:** 6
**Confidence:** 4

**Summary:**

This paper proposes a novel neural network model, parallel-structured and all-component Fourier neural operator (PAC-FNO) to address visual recognition under low-quality images. By operating in the frequency domain, PAC-FNO is able to learn the semantics of images in various resolutions and/or natural variations for challenging image recognition with a single model. The proposed PAC-FNO is capable of handling both low-resolution and input variations typically observed in low-quality images with a single model. It can also be attached to a downstream visual recognition model, which is beneficial for handling multiple input variations at once and minimizing the changes in the downstream model during fine-tuning. In the evaluation with four visual recognition models and seven datasets, the proposed PAC-FNO achieves excellent performance.

**Strengths:**

1. The paper is organized well.
2. Extensive experimental results are provided to illustrate the effectiveness of the proposed method.

**Weaknesses:**

1. Is there a more advanced choice for the SR baseline model used for comparison in your experimental setup? This will affect the fairness of the performance of your experiment?
2. It can be found that in ViT-B16, PAC-FNO shows not very good results at all low resolutions compared to other methods. What caused this phenomenon to occur? Is your method also unfriendly to other Transformer methods?
3. The ideal low-pass filter in the FNO block removes detailed image signals that play an important role in classification in the fine-grained dataset. Is this conclusion applicable to Transformer based image classification methods? More quantitative results should be provided to confirm the universality of the proposed method.
4. The ablation experiments about the results of the zero-padding operation and the exclusion of the low pass filter need to be completed to explain the design of the AC-FNO block.

**Questions:**

Please see the weaknesses above.

---

> ### Author Response · Authors · 2023-11-17
>
> Thank you for the encouraging remarks about our paper’s extensive experimental results and organizing and the valuable feedback from the reviewer. We hope that our responses will solve the reviewer’s questions.
>
> > 1. Is there a more advanced choice for the SR baseline model used for comparison in your experimental setup? This will affect the fairness of the performance of your experiment?
>
> $\to$ For fairness of experiment in SR baseline, we will report additional baseline equipped with the latest super-resolution model in Table 13 in Appendix F.3 of the revision that will be updated shortly around November 20th AoE.
>
> > 2. It can be found that in ViT-B16, PAC-FNO shows not very good results at all low resolutions compared to other methods. What caused this phenomenon to occur? Is your method also unfriendly to other Transformer methods?
>
> $\to$ PAC-FNO generally performs well in low resolution but only worse than A-FNO. A-FNO is a model proposed as a token mixer for transformers, so it seems particularly friendly when combined with Transformer-based models.
>
> > 3. The ideal low-pass filter in the FNO block removes detailed image signals that play an important role in classification in the fine-grained dataset. Is this conclusion applicable to Transformer based image classification methods? More quantitative results should be provided to confirm the universality of the proposed method.
>
> $\to$ Yes, high-frequency components play a more important role in image classification, especially fine-grained datasets. So, to see the effect of high frequency component removal for nuanced classification problem, we conduct  a fine-grained classification experiment with the ViT model. On the Oxford-IIIT Pets dataset, PAC-FNO, which captures the high-frequency components, performs better at all resolutions than FNO with an ideal low-pass filter.compared to FNO with an ideal low-pass filter. We add these results in Tables 16 and 17 in Appendix F.4 of the revision.
>
> | Oxford-IIIT Pets |  28  |  32  |  56  |  64  |  112 |  128 |  224 |
> |:----------------:|:----:|:----:|:----:|:----:|:----:|:----:|:----:|
> |        FNO       | 26.3 | 33.0 | 58.1 | 64.8 | 82.9 | 85.8 | 91.3 |
> |      PAC-FNO     | 40.3 | 46.2 | 69.0 | 72.5 | 86.8 | 89.2 | 92.2 |
>
> > 4. The ablation experiments about the results of the zero-padding operation and the exclusion of the low pass filter need to be completed to explain the design of the AC-FNO block.
>
> $\to$ Zero-padding works to upscale low-resolution images that are smaller than the target resolution in the frequency domain to the target resolution. In other words, if there is no zero-padding operation, low-resolution images cannot be processed.
> Then, we provide the ablation experiments for low-pass filters in the first row in Table 21 in Appendix F.7 of the revision. When a low-pass filter is used, performance decreases in terms of accuracy because high-frequency components cannot be considered, but the performance decrease is large for input variation.
>
> |           ImageNet-1k           |  32  |  64  |  128 |  224 |
> |:-------------------------------:|:----:|:----:|:----:|:----:|
> |  PAC-FNO (low pass filter) | 53.5 | 71.4 | 78.7 | 79.0 |
> |             PAC-FNO             | 58.9 | 74.5 | 80.2 | 81.5 |
>
> |         ImageNet-C/P Fog        |  32  |  64  |  128 |  224 |
> |:-------------------------------:|:----:|:----:|:----:|:----:|
> |  PAC-FNO (low pass filter) | 18.0 | 41.7 | 52.4 | 54.4 |
> |             PAC-FNO             | 25.4 | 48.2 | 60.1 | 62.8 |

---

> ### Author Response · Authors · 2023-11-20
>
> We provide an additional comparison to the method equipped with the latest super-resolution model. The results are in the following table. OSRT [1] is a state-of-the-art model in the super-resolution domain but it does not support $\times$8 upscale. Therefore, we only use the $\times$2 and $\times$4 upscale models of OSRT. In addition, we report OSRT (fine-tune) that combines super-resolution and fine-tuning methods. That is, we fine-tune the pre-trained classification model with low-resolution images upscaled by a super-resolution model.  As a result, PAC-FNO shows better performance than OSRT and OSRT (fine-tune) methods. We also report these results in Table 13 in Appendix F.3 of the revision.
>
> |  Imagenet-1k  |    Method   |        Metric       |    28    |    32    |    56    |    64    |    112   |    128   |  224 |
> |:-------------:|:-----------:|:-------------------:|:--------:|:--------:|:--------:|:--------:|:--------:|:--------:|:----:|
> |              |     OSRT    |    Top-1 Acc (%)    |     -    |     -    |   61.4   |     -    |   75.4   |     -    | 82.5 |
> |               |             | # of Parameters (M) |     -    |     -    |   11.93  |     -    |   11.93  |     -    |   -  |
> |  ConvNeXt-Tiny  |     OSRT    |    Top-1 Acc (%)    |     -    |     -    |   71.2   |     -    |   78.4   |     -    | 82.1 |
> |               | (Fine-tune) | # of Parameters (M) |     -    |     -    |   11.93  |     -    |   11.79  |     -    |   -  |
> |               |   PAC-FNO   |    Top-1 Acc (%)    |  **58.9**   | **63.2** | **77.6** | **76.2** | **80.2** | **80.7** | 81.8 |
> |              |             | # of Parameters (M) |          |          |          |   3.65   |          |          |      |
>
> | Imagenet-C/P Fog |    Method   |        Metric       |    28    |    32    |    56    |    64    |    112   |    128   |    224   |
> |:----------------:|:-----------:|:-------------------:|:--------:|:--------:|:--------:|:--------:|:--------:|:--------:|:--------:|
> |                 |     OSRT    |    Top-1 Acc (%)    |     -    |     -    |   19.4   |     -    |   37.9   |     -    |   58.4   |
> |                  |             | # of Parameters (M) |     -    |     -    |   11.93  |     -    |   11.93  |     -    |     -    |
> | ConvNeXt-Tiny  |     OSRT    |    Top-1 Acc (%)    |     -    |     -    |   42.3   |     -    |   56.4   |     -    |   59.4   |
> |                  | (Fine-tune) | # of Parameters (M) |     -    |     -    |   11.93  |     -    |   11.79  |     -    |     -    |
> |                  |   PAC-FNO   |    Top-1 Acc (%)    | **25.4** | **30.4** | **48.2** | **51.7** | **60.1** | **61.4** | **62.8** |
> |         `        |             | # of Parameters (M) |          |          |          |   3.65   |          |          |          |
>
> [1] Yu, Fanghua, et al. "OSRT: Omnidirectional image super-resolution with distortion-aware transformer." Proceedings of the IEEE/CVF Conference on Computer Vision and Pattern Recognition. 2023.

---

> ### Author Response · Authors · 2023-11-22
>
> Dear Reviewer WCy8,
>
> We appreciate the reviewer’s time and effort in reviewing our manuscript and insightful comments.
>
> As the closure of the discussion period is approaching, we would like to bring the review’s attention and check if the reviewer could let us know whether the concerns or the misunderstanding have been addressed.
>
> If this is the case, we would appreciate if you could adjust your rating accordingly.
>
> Best regards,
>
> Authors

---

> > ### Comment · Reviewer_WCy8 · 2023-11-22
> >
> > Thanks for your time! My major concerns have already been addressed. I keep my original score.

---

### Official Review · Reviewer_nwGj · 2023-11-02

**Soundness:** 3 good
**Presentation:** 3 good
**Contribution:** 3 good
**Rating:** 6
**Confidence:** 4

**Summary:**

This work has developed a neural network architecture for image recognition that is designed to address the influence of complex degradation factors. It aims to capture both low-frequency and high-frequency components to balance accuracy and generalization. The authors first propose to discard the low-pass filters in the existing FNO structure to retain all frequency components. Subsequently, a parallel structure is introduced to further enhance the utilization of frequency domain information. Finally, the authors design a two-stage training strategy to ensure performance stability.

**Strengths:**

1. The overall paper has a clear logical structure, and the explanation of the methodology and the presentation of the constructed mechanisms are intuitive and easy to understand.
2. The author provides a sufficiently detailed explanation for the motivation behind each component in PAC-FNO.
3. The problem that this work aims to address holds a certain degree of practical application value.

**Weaknesses:**

1. The abandonment of the low-pass filter is one of the main innovations in this work. Although the author provides an explanation for the motivation behind this operation, it is still recommended that the author conduct ablative experiments to analyze the impact of low-frequency/high-frequency information on accuracy/generalization.
2. As for parallel architecture, the relevant experimental results have indeed proven its effectiveness. However, the explanation of parallel architecture in the method section appears somewhat lacking. It is hoped that the author can provide further analysis of the mechanism that enables it to be effective.
3. In terms of comparative experiments, the methods used by the author for comparison appear to be lacking in both quantity and novelty. The comprehensiveness of the complex scenarios considered by the author is commendable, but it is hoped that the author can still increase the comparison results with more advanced works to more effectively validate the superiority of the proposed method.
4. The author mentions the advantages of this work in terms of efficiency, but it seems that no experimental analysis related to efficiency has been provided (such as FLOPs and runtime on data at different resolutions).

**Questions:**

Please refer to the Weaknesses.

---

> ### Author Response · Authors · 2023-11-17
>
> Thank you for the encouraging remarks about our paper’s clear methodology explanation and the valuable feedback from the reviewer. We hope that our responses will solve the reviewer’ questions.
>
> > 1. The abandonment of the low-pass filter is one of the main innovations in this work. Although the author provides an explanation for the motivation behind this operation, it is still recommended that the author conduct ablative experiments to analyze the impact of low-frequency/high-frequency information on accuracy/generalization.
>
> $\to$ Thank you for the suggestion! We provide an analysis of the impact of low and high-frequency information on accuracy/generalization through ablation experiments in Table 21 in Appendix F.7 of revision. Compared to PAC-FNO, accuracy and generalization decrease when using low-pass filter or high-pass filter. PAC-FNO with low pass filter show similar performance in ImageNet-1k compared with our PAC-FNO, but show a decrease in performance in terms of generalization in ImageNet-C/P Fog. On the other hand, when using a high-pass filter, it is expected to show good performance in ImageNet-C/P Fog, but it does not show good performance in terms of generalization because the performance is also poor in ImageNet-1k. Therefore, PAC-FNO, which uses both low-frequency and high-frequency components, only shows good performance in terms of accuracy/generalization.
>
> |           ImageNet-1k           |  32  |  64  |  128 |  224 |
> |:-------------------------------:|:----:|:----:|:----:|:----:|
> |  PAC-FNO (low pass filter) | 53.5 | 71.4 | 78.7 | 79.0 |
> | PAC-FNO (high pass filter) | 21.6 | 49.4 | 68.2 | 74.8 |
> |             PAC-FNO             | 58.9 | 74.5 | 80.2 | 81.5 |
>
> |         ImageNet-C/P Fog        |  32  |  64  |  128 |  224 |
> |:-------------------------------:|:----:|:----:|:----:|:----:|
> |  PAC-FNO (low pass filter) | 18.0 | 41.7 | 52.4 | 54.4 |
> | PAC-FNO (high pass filter) |  5.92  | 23.0 | 43.2 | 50.2 |
> |             PAC-FNO             | 25.4 | 48.2 | 60.1 | 62.8 |
>
> > 2. As for parallel architecture, the relevant experimental results have indeed proven its effectiveness. However, the explanation of parallel architecture in the method section appears somewhat lacking. It is hoped that the author can provide further analysis of the mechanism that enables it to be effective.
>
> $\to$ The reason that it is effective is that the parallel configuration captures both low and high-frequency components while the serial architecture captures low-frequency components only (first paragraph in Sec. 3.2). Both low and high-frequency components must be captured to achieve good accuracy/generalization. This can be confirmed in Table 21 in Appendix F.7 in the revision. We visualize what frequency the parallel configuration and serial configuration capture in Figure 7 in Appendix F.8 of revision. Figure 7 shows that the parallel configuration more captures high frequencies than the serial configuration.

---

> ### Author Response · Authors · 2023-11-17
>
> > 3. In terms of comparative experiments, the methods used by the author for comparison appear to be lacking in both quantity and novelty. The comprehensiveness of the complex scenarios considered by the author is commendable, but it is hoped that the author can still increase the comparison results with more advanced works to more effectively validate the superiority of the proposed method.
>
> $\to$ We believe that advanced work to effectively validate the superiority of our method is an advanced super-resolution method, as super-resolution models are relatively old models. Therefore, we provide additional comparison to the method equipped with the latest super-resolution model in Appendix F.4 of the revision that will be updated shortly around November 20th AoE. Furthermore, we report additional baselines that combine super-resolution and fine-tuning methods in rows 5-6 in Table 13 in Appendix F.4 of the revised paper. That is, we fine-tune the pre-trained classification model with low-resolution images upscaled by a super-resolution model. This method showed worse performance than our PAC-FNO.
> In addition, super-resolution methods are needed for each resolution. In other words, upscaling models of x8, x4, and x2 are needed to handle 28, 56, and 112 resolution, respectively. In contrast, our proposed PAC-FNO can handle images of all resolutions with an additional 3.65M network and shows good performance.
> If the advanced works you intend are not super-resolution models, please let us know.
>
> > 4. The author mentions the advantages of this work in terms of efficiency, but it seems that no experimental analysis related to efficiency has been provided (such as FLOPs and runtime on data at different resolutions).
>
> $\to$ We did not mention the efficiency as the advantage of this work. Instead, we stated "efficacy of our neural operator-based mechanism" (not the “efficiency”) in the last paragraph of Section 4.4 is that neural operator-based mechanism is suitable for real-world applications because it can handle a variety of resolutions without the process of resizing to the target resolution. But for curiosity, we also compare the efficiency of our method to the other methods in the following table:
>
> | ImageNet-1k |    Metrics   |    28   |   56   |   112   |  224  |
> |:-----------:|:------------:|:-------:|:------:|:-------:|:-----:|
> |    Resize   |    GFLOPs    |   8.96  |  8.96  |   8.96  |  8.96 |
> |             | Runtimes (s) |  0.006  |  0.006 |  0.006  | 0.006 |
> |  Fine-tune  |    GFLOPs    |   8.96  |  8.96  |   8.96  |  8.96 |
> |             | Runtimes (s) |  0.006  |  0.006 |  0.006  | 0.006 |
> |     DRLN    |    GFLOPs    |  180.66 | 412.05 | 1200.50 |  8.96 |
> |             | Runtimes (s) |  0.378  |  0.498 |  0.913  | 0.007 |
> |     DRPN    |    GFLOPs    | 1220.42 | 576.94 |  387.88 |  8.96 |
> |             | Runtimes (s) |  0.1532 |  0.164 |  0.171  | 0.007 |
> |     FNO     |    GFLOPs    |   9.78  |  9.78  |   9.78  |  9.78 |
> |             | Runtimes (s) |  0.016  |  0.016 |  0.016  | 0.016 |
> |     UNO     |    GFLOPs    |   9.10  |  9.10  |   9.10  |  9.10 |
> |             | Runtimes (s) |  0.018  |  0.018 |  0.018  | 0.018 |
> |     AFNO    |    GFLOPs    |   8.96  |  8.96  |   8.96  |  8.96 |
> |             | Runtimes (s) |  0.010  |  0.010 |  0.010  | 0.010 |
> |   PAC-FNO   |    GFLOPs    |   8.98  |  8.98  |   8.98  |  8.98 |
> |             | Runtimes (s) |  0.013  |  0.013 |  0.013  | 0.013 |
>
> As a result, PAC-FNO showed the most efficiency in terms of FLOPs and runtimes except AFNO.

---

> ### Author Response · Authors · 2023-11-20
>
> We provide an additional comparison to the method equipped with the latest super-resolution model. The results are in the following table. OSRT [1] is a state-of-the-art model in the super-resolution domain but it does not support $\times$8 upscale. Therefore, we only use the $\times$2 and $\times$4 upscale models of OSRT. As a result, PAC-FNO shows better performance than OSRT and OSRT (fine-tune) methods. We also report these results in Table 13 in Appendix F.3 of the revision.
>
> |  Imagenet-1k  |    Method   |        Metric       |    28    |    32    |    56    |    64    |    112   |    128   |  224 |
> |:-------------:|:-----------:|:-------------------:|:--------:|:--------:|:--------:|:--------:|:--------:|:--------:|:----:|
> |               |     DBPN    |    Top-1 Acc (%)    |   40.7   |     -    |   68.2   |     -    |   79.4   |     -    | 82.5 |
> |               |             | # of Parameters (M) |   23.21  |     -    |   10.43  |     -    |   5.95   |     -    |   -  |
> |               |     DBPN    |    Top-1 Acc (%)    |   **60.8**   |     -    |   72.5   |     -    |   76.7   |     -    | 82.5 |
> |               | (Fine-tune) | # of Parameters (M) | 23.21 |     -    |   10.43   |     -    |   5.95   |     -    | - |
> | ConvNeXt-Tiny |     OSRT    |    Top-1 Acc (%)    |     -    |     -    |   61.4   |     -    |   75.4   |     -    | 82.5 |
> |               |             | # of Parameters (M) |     -    |     -    |   11.93  |     -    |   11.93  |     -    |   -  |
> |               |     OSRT    |    Top-1 Acc (%)    |     -    |     -    |   71.2   |     -    |   78.4   |     -    | 82.1 |
> |               | (Fine-tune) | # of Parameters (M) |     -    |     -    |   11.93  |     -    |   11.79  |     -    |   -  |
> |               |   PAC-FNO   |    Top-1 Acc (%)    |   58.9   | **63.2** | **77.6** | **76.2** | **80.2** | **80.7** | 81.8 |
> |              |             | # of Parameters (M) |          |          |          |   3.65   |          |          |      |
>
> | Imagenet-C/P Fog |    Method   |        Metric       |    28    |    32    |    56    |    64    |    112   |    128   |    224   |
> |:----------------:|:-----------:|:-------------------:|:--------:|:--------:|:--------:|:--------:|:--------:|:--------:|:--------:|
> |                  |     DBPN    |    Top-1 Acc (%)    |   0.67   |     -    |   0.99   |     -    |   1.32   |     -    |   58.4   |
> |                  |             | # of Parameters (M) |   23.21  |     -    |   10.43  |     -    |   5.95   |     -    |     -    |
> |                  |     DBPN    |    Top-1 Acc (%)    |   21.8   |     -    |   42.3   |     -    |   56.8   |     -    |   61.0   |
> |                  | (Fine-tune) | # of Parameters (M) |     -    |     -    |   10.43  |     -    |   5.95   |     -    |     -    |
> |   ConvNeXt-Tiny  |     OSRT    |    Top-1 Acc (%)    |     -    |     -    |   19.4   |     -    |   37.9   |     -    |   58.4   |
> |                  |             | # of Parameters (M) |     -    |     -    |   11.93  |     -    |   11.93  |     -    |     -    |
> |                  |     OSRT    |    Top-1 Acc (%)    |     -    |     -    |   42.3   |     -    |   56.4   |     -    |   59.4   |
> |                  | (Fine-tune) | # of Parameters (M) |     -    |     -    |   11.93  |     -    |   11.79  |     -    |     -    |
> |                  |   PAC-FNO   |    Top-1 Acc (%)    | **25.4** | **30.4** | **48.2** | **51.7** | **60.1** | **61.4** | **62.8** |
> |         `        |             | # of Parameters (M) |          |          |          |   3.65   |          |          |
>
>
> [1] Yu, Fanghua, et al. "OSRT: Omnidirectional image super-resolution with distortion-aware transformer." Proceedings of the IEEE/CVF Conference on Computer Vision and Pattern Recognition. 2023.

---

> ### Author Response · Authors · 2023-11-22
>
> Dear Reviewer nwGj,
>
> We appreciate the reviewer’s time and effort in reviewing our manuscript and insightful comments.
>
> As the closure of the discussion period is approaching, we would like to bring the review’s attention and check if the reviewer could let us know whether the concerns or the misunderstanding have been addressed.
>
> If this is the case, we would appreciate if you could adjust your rating accordingly.
>
> Best regards,
>
> Authors

---

### Author Response · Authors · 2023-11-17

Dear All Reviewers,

We thank the reviewers for taking the time to read, evaluate, and provide valuable feedback.
We upload a rebuttal revision that includes feedback from reviewers. Responses to reviewers feedback are highlighted in blue.
Results for the state of the art super resolution model will be updated shortly around November 20th AoE.

Best regards,
Authors

---

### Author Response · Authors · 2023-11-20

Dear All Reviewers,

We uploaded a new version of the revision which contains results for the state-of-the-art model in the super-resolution domain. We revised the following points and uploaded a new version:
1. We added additional experimental results of the super-resolution method in Table 13 in Appendix F.3.
2. We added experimental results on fine-grained datasets in ViT models in Tables 16 and 17  in Appendix F.4.
3. We added additional ablation studies results of the performance of PAC-FNO according to low and high-frequency filters in Table 21 in Appendix F.7.
4. We describe the effectiveness of parallel architecture and how it affects input variation in terms of frequency in Figure 7 and Figure 8 in Appendix F.8.
5. We added FLOPs and runtime on data at different resolutions in Table 22 in Appendix F.9.

Best regards,
Authors

---

### Meta-Review · Area_Chair_E9pe · 2023-12-05

**Metareview:**

All three reviewers have given a final rating of 6, indicating a leaning towards accepting the work. After carefully examining the reviewers' comments and the responses provided by the authors, it is clear that the reviewers appreciate the professionalism in the presentation and the solid experimental validation of the work. Additionally, after the authors provided rebuttals, two reviewers mentioned that the authors effectively addressed their concerns, and one of them even upgraded their rating, acknowledging the contributions of the work. In summary, the work has demonstrated strength in experimental validation, motivation exposition, and method explanation, receiving recognition from the majority of the reviewers. The authors also provided sufficiently convincing explanations during the response phase. Therefore, I decide to accept this work.

**Justification For Why Not Higher Score:**

Please refer to the metareview.

**Justification For Why Not Lower Score:**

Please refer to the metareview.

---

### Decision · Program_Chairs · 2024-01-16

Accept (poster)